# Urban centre green metrics in Great Britain: A geospatial and socioecological study

Jake M. Robinson[1,2,3]*, Suzanne Mavoa[4,5], Kate Robinson[6], Paul Brindley[1]*

1 Department of Landscape Architecture, Arts Tower, University of Sheffield, Sheffield, United Kingdom,
2 College of Science and Engineering, Flinders University, Adelaide, South Australia, Australia, 3 *in*VIVO Planetary Health, of the Worldwide Universities Network (WUN), West New York, NJ, United States of America, 4 Melbourne School of Population and Global Health, University of Melbourne, Melbourne, VIC, Australia, 5 Environmental Public Health Branch, Environment Protection Authority Victoria, Melbourne, VIC, Australia, 6 Data Insight Team, Arts Tower, University of Sheffield, Sheffield, United Kingdom

* bio.jmr@gmail.com (JMR); p.brindley@sheffield.ac.uk (PB)

## Abstract

Green infrastructure plays a vital role in urban ecosystems. This includes sustaining biodiversity and human health. Despite a large number of studies investigating greenspace disparities in suburban areas, no known studies have compared the green attributes (e.g., trees, greenness, and greenspaces) of urban *centres*. Consequently, there may be uncharacterised socioecological disparities between the cores of urban areas (e.g., city centres). This is important because people spend considerable time in urban centres due to employment, retail and leisure opportunities. Therefore, the availability of—and disparities in—green infrastructure in urban centres can affect many lives and potentially underscore a socio-ecological justice issue. To facilitate comparisons between urban centres in Great Britain, we analysed open data of urban centre boundaries with a central business district and population of $\geq$100,000 ($n = 68$). Given the various elements that contribute to 'greenness', we combine a range of different measurements (trees, greenness, and accessible greenspaces) into a single indicator. We applied the normalised difference vegetation index (NDVI) to estimate the mean greenness of urban centres and the wider urban area (using a 1 km buffer) and determined the proportion of publicly accessible greenspace within each urban centre with Ordnance Survey Open Greenspace data. Finally, we applied a land cover classification algorithm using i-Tree Canopy to estimate tree coverage. This is the first study to define and rank urban centres based on multiple green attributes. The results suggest important differences in the proportion of green attributes between urban centres. For instance, Exeter scored the highest with a mean NDVI of 0.15, a tree coverage of 11.67%, and an OS Greenspace coverage of 0.05%, and Glasgow the lowest with a mean NDVI of 0.02, a tree cover of 1.95% and an OS Greenspace coverage of 0.00%. We also demonstrated that population size negatively associated with greenness and tree coverage, but not greenspaces, and that green attributes negatively associated with deprivation. This is important because it suggests that health-promoting and biodiversity-supporting resources diminish as population and deprivation increase. Disparities in green infrastructure across the country, along with the population and deprivation-associated trends, are important in terms of socioecological and equity justice. This study provides a baseline and stimulus to

**Data Availability Statement:** All relevant data are within the manuscript and its Supporting Information files.

**Funding:** The funding received for this manuscript is as follows: University of Melbourne Faculty of

Medicine, Dentistry and Health Sciences Research Fellowship., Dr Suzanne Mavoa University of Sheffield, Open Access publishing agreement with PLOS ONE, Dr Jake M. Robinson and Dr. Paul Brindley.

**Competing interests:** The authors have declared that no competing interests exist.

help local authorities and urban planners create and monitor equitable greening interventions in urban/city centres.

## Introduction

It is projected that nearly 70% of the world's population will be living in towns and cities by 2050 [1]. The process of urbanisation places considerable pressure on biodiversity and human health [2]; for example, by degrading habitats and increasing harmful pollutants such as gases (e.g., nitrogen dioxide, sulphur dioxide, ozone) and particulate matter [3].

Evidence shows that green infrastructure, including trees, hedgerows, green roofs, and parks, plays a vital role in urban ecosystem integrity [4, 5]. This includes sustaining biodiversity. For example, many animals rely upon the resources associated with semi-natural habitats (e.g. vegetation communities provide nutrition and refuge for invertebrates, birds, mammals, reptiles and amphibians). Green infrastructure can provide habitat corridors and connections to the broader landscape, allowing animals, plants, and microbes to disperse and exchange genes [6, 7]. Additionally, green infrastructure provides a range of human health and wellbeing benefits. Indeed, street trees can reduce the negative health impacts associated with the urban heat island effect, and hedgerows can act as pollution barriers [8, 9]. Greenspaces provide supportive environments for recreation [10], conviviality, and creativity [11]. These urban green attributes can also reduce stress and anxiety [12], provide positive affect [e.g., 13], and potentially help regulate the immune system via interactions with environmental microbiota [14, 15].

Studies assessing the presence and impacts of urban green attributes typically focus on the places where most people reside, such as suburban zones [16–18]. Several studies have also used remote sensing-based green cover identification methods in cities. For instance, NDVI is widely used to estimate green cover in urban areas [19, 20], and the Enhanced Vegetation Index (EVI) has also been used [21]. NDVI is considered more sensitive to leaf chlorophyll concentrations via the red spectral band (620–670 nm), while EVI more sensitive to canopy structure and leaf area via the near-infrared (NIR) band (840–875 nm) [22]. Ordnance Survey's (Britain's national mapping agency) Open Greenspace data is also widely used in urban greenspace studies to explore the distribution of publicly accessible greenspaces [23, 24]. However, no known studies have comparatively assessed the urban green attributes of urban/city *centres* using multiple metrics–although studies have assessed individual green attributes across the wider suburban area [e.g., 25, 26].

Many people from diverse backgrounds spend considerable time (e.g. for employment, shopping, and recreation) in urban centres [27]. Therefore, urban centres are places where populations from otherwise socioeconomically disparate areas merge and mingle. However, little is known about the equity of green infrastructure provision in city centres. Disparities in this infrastructure could underscore an important socio-ecological justice issue, with some populations gaining the benefits of healthy urban ecosystems and others enduring the disbenefits of poor green infrastructure provision. The same applies to biodiversity––i.e., it is important to understand potential disparities in wildlife-supporting habitats in city centres. Considering these factors, we argue that more emphasis should be placed on mapping and enhancing the green attributes (e.g., trees, broader vegetation cover, and publicly accessible greenspaces such as parks) of urban cores/centres. In doing so, there is considerable potential to provide a range of positive health benefits to many people across the socioeconomic spectrum via the augmented provision of health-promoting green features such as urban forests

and recreational greenspaces. There is also potential to enhance biodiversity and interspecies health.

Urban centres can be challenging to define due to being complex socioeconomic systems that evolve with expanding and contracting spatial and compositional extents [28–30]. Past research has defined urban centres based on land-use plans [27] or postcodes [31]. Yet, there is no consistent and robust dataset for GB-wide urban centres using these approaches. However, we identified an established, standardised approach using the Consumer Data Research Centre's (CDRC) national-level geodata packs [32].

### Objectives

The main objectives of this study were to: **(a)** define urban centres in Great Britain (GB) (Northern Ireland was excluded due to data unavailability); **(b)** map and characterise the green attributes of urban centres in Great Britain (based on three different metrics for robustness: greenness as defined by the NDVI—a remote sensing metric; tree coverage; and publicly accessible greenspaces such as parks and sports fields), and provide a table of rankings to establish vital baseline data; **(c)** determine whether the level of greenness within urban centres is reflected across the wider urban area (1 km radius); **(d)** determine whether there is a relationship between the size of the urban area (as a whole) and level of green attributes within the urban centre, and **(e)** determine whether there is a relationship between relative deprivation and level of green attributes within urban centres. These objectives allowed us to map and understand potential disparities in green infrastructure provision in Great British urban centres. To achieve these aims, we applied a range of geospatial methods, including the manipulation of data from a density spatial clustering of applications with noise (DBSCAN), along with the normalised difference vegetation index (NDVI), the i-Tree Canopy land classification algorithm, and OS Open Greenspace data.

## Materials and methods

### GB urban centre boundaries

To define the boundaries of GB urban centres (Fig 1), we used the CDRC's national-level geodata packs to acquire boundaries and centroids for retail centres [32]. The data were produced from the 2015 local data company's (LDC) retail units location dataset. CRDC built these retail centre boundaries using the Graph-DBSCAN clustering method [29]. This method implements a sparse graph representation of retail unit locations based on a distance-constrained k-nearest neighbour adjacency list that is decomposed using the Depth First Search algorithm [29].

We used the Retail Centre Typology (2018) linked with the CDRC boundaries. This multi-dimensional taxonomy of retail and consumption spaces focuses on four domains: (1) the composition of spaces; (2) the diversity of spaces; (3) the function of spaces; and (4) the economic vitality of the centres [32]. We used filters to select the CDRC retail centre typology "*premium retail and leisure destinations of semi-regional importance*". This typology is classed as the highest level regional urban centre based on the above criteria. The other inclusion criterion was the selection of retail boundaries within settlements with a population size of at least 100,000. This helped to reduce highly skewed comparative scenarios, e.g. comparing the small city of St David's in Wales (population size: <1,500) with Birmingham in England (population size: >1,000,000). We used QGIS version 3.4 [33] and ArcGIS version 10.8 [34] to import the CDRC.gpkg file and extract the retail centre boundary layers by using in-built algorithms, creating new feature layers (geopackageFeatureTable). QGIS was used to create shapefiles from these boundary features in new vector layers (Layer > New > New Shapefile Layer) and to

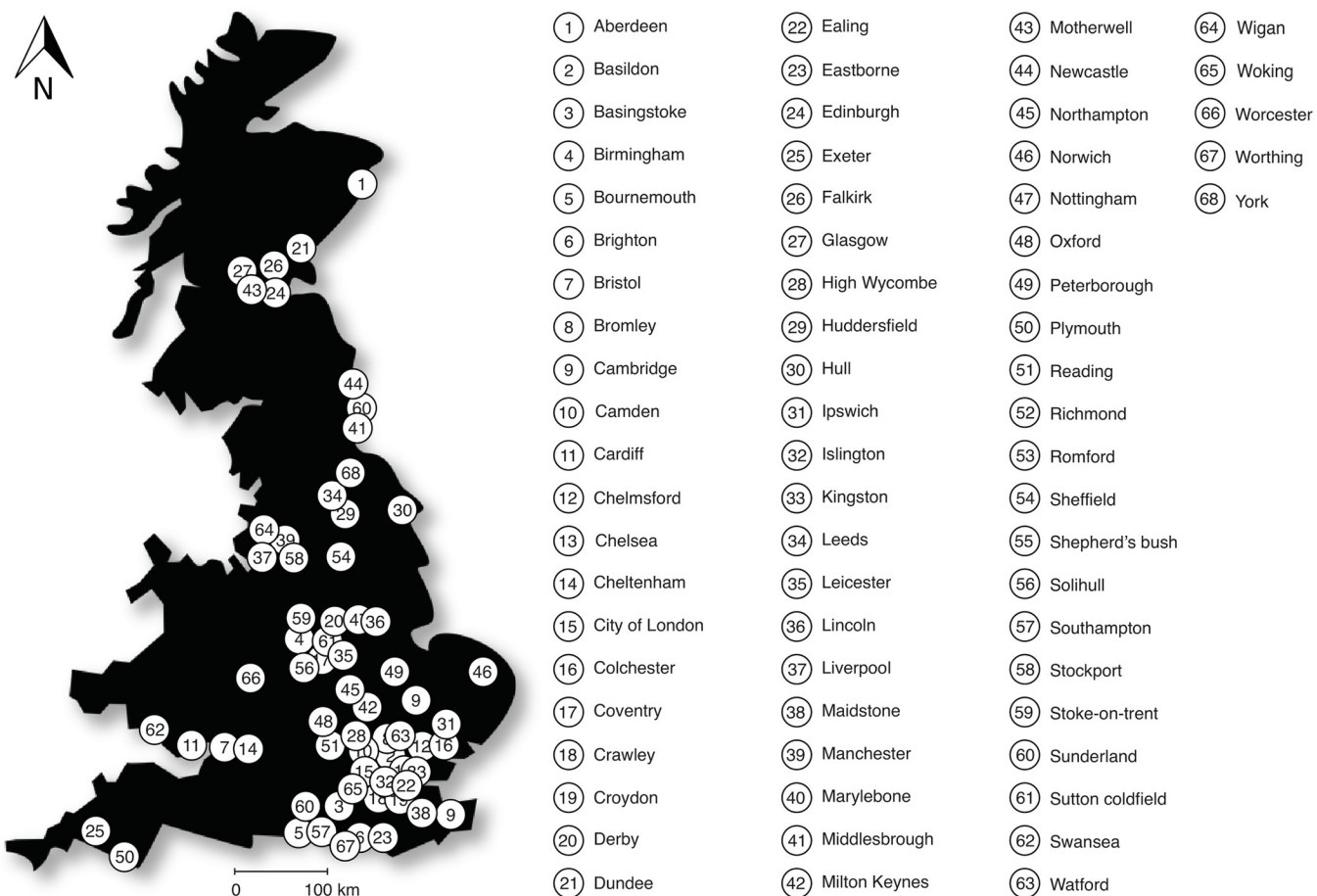

**Fig 1. Distribution of the 68 GB urban centres.** These comprise 59 in England, 3 in Wales, and 6 in Scotland. Urban centres are listed in alphabetical order. The outline of the GB map was from www.pixabay.com and used under a CC-BY-4.0 licence.

process the geospatial data. We acquired population data from the Office for National Statistics NOMIS web portal [35]. Population estimates were based on 2011 Census data (which uses OS Built-Up Area boundaries; [36]) and were downloaded as a.csv file and integrated into the urban centre attribute tables in ArcGIS via the Add Join algorithm (Data Management). We excluded urban centres in Northern Ireland due to a lack of appropriate spatial data to define urban boundaries and greenspaces.

## Mean greenness

To estimate the mean greenness of each GB urban centre, we acquired the Copernicus Sentinel-2 atmospherically-corrected satellite imagery (10 m resolution). The Sentinel-2 satellite collected this dataset in 2019, and it was downloaded by the researchers from the EDINA Digimap Service in August 2020 [37]. The Sentinel-2 images used were cloud-free composites collected on various dates and sourced across the calendar year 2019. We acquired spectral bands 4 (Red) and 8 (Near Infrared) and applied the Normalised Difference Vegetation Index equation as follows:

$$\text{NDVI} = \frac{Near\ Infrared - Red}{Near\ Infrared + Red}$$

This equation provides a score of between -1 to 1. The score provides an estimation of land-cover greenness (a proxy for chlorophyll output), where -1 represents a very low level of green-ness and 1 represents a very high level of greenness. This 'greenness' score has been used as a proxy for vegetation biomass and vegetation cover in other green infrastructure and geospatial studies [38, 39], hence being considered suitable for this study (whilst recognising other indices are available). In QGIS, we created an algebraic expression to process the raster (.tif) files, i.e. the two Sentinel-2 spectral band layers: red and near-infrared using the algebraic expression calculator. Using the zonal statistics raster analysis tool, we calculated the mean NDVI values (with negative values removed as they may represent water bodies) for all areas within the predefined GB urban centre boundaries. We also calculated the mean NDVI values of radial buffers spanning 1 km from the urban centre boundaries (e.g., by importing the Sentinel 2 files, creating new polygon layers and using the Raster Calculator expressions: applying the NDVI formula ("band 8"–"band 4") / ("band 8" + "band 4") for each tile and saving as.tif). The urban centre boundaries were clipped out in ArcGIS using a cookie-cutter approach (via Vector > Geoprocessing tools > Clip). This allows us to compare an urban centre to its context and potentially account for any residual bias remaining from the standardisation of NDVI values across the country.

## Tree canopy coverage

To estimate tree canopy coverage in the urban centres, we used the land classification algorithm tool i-Tree Canopy [40], which has been used in previous urban greenspace studies [41–43]. The urban centre boundaries were loaded into i-Tree Canopy. This web-based tool enabled random sampling points (at least 300 points per boundary) and selection of tree canopy cover metrics, which is overlaid with Landsat 8 2020 satellite imagery. The i-Tree Canopy tool provides a graphical and map output with land cover classification metrics, including % cover and area (ha) with standard error. For city-level assessments such as our urban centre boundaries, 300 random points with a standard error of <2% are recommended [34]. These conditions were met for each of the 68 urban boundaries in this study.

## Greenspace attributes

To determine greenspace presence, type, number of greenspaces, and area in each GB urban centre, we downloaded and imported the OS Open Greenspace dataset (data version April 2020) [44]. This dataset contains data for publicly accessible urban greenspaces in GB. Before analysis, we removed all greenspaces classified as sports/leisure facilities due to the high level of grey space (e.g. buildings such as leisure centres and carparks) and a low level of greenspace. See S1 Table for a full breakdown of OS Greenspaces in each urban centre. Using vector geoprocessing tools and the field calculator within QGIS, we calculated metrics on the abundance and area ($m^2$) of publicly accessible OS Greenspaces that occurred within each of the 68 urban centre boundary layers. Specifically, we used the Intersection Tool with both layers as input layers. This resulted in a layer which contains every polygon inside the urban centre boundary, and the attribute table contains all attributes from both input layers, including a number of polygons. Using the field calculator, the '$area' operator was used to calculate the area of each urban centre boundary.

## Relative deprivation

To assess whether relative deprivation was associated with the level of green attributes in urban centres, a measure of deprivation was generated for each urban area using the 2019 local authority district summaries Index of Multiple Deprivation (IMD; [45]). Three urban centres

fell across multiple local authority districts. In these cases, weighting of deprivation was under-taken based on the area of the district. Area, as opposed to population, was used for the weight as the focus of this work—urban centres—are inherently a blend of both populated and non-populated geographic areas. The IMD provides an output of relative deprivation based on a multivariate analysis of demographic data such as crime risk, health, economics, living envi-ronment, and education. Deprivation could only be tied to the 60 English urban centres as dif-ferent incompatible data exist for the other countries of Great Britain.

### Ranking and statistical analysis

Statistical analysis was carried out in R and SPSS, with supplementary software including Microsoft Excel (for.csv file processing and constructing the parallel coordinate plot). To assess potential relationships between the green attributes (tree cover, greenness, OS Greenspaces) with population size, deprivation and urban centre area, we used the non-parametric Spear-man's Rank Correlation Coefficient test in R. This was because the data were non-normally distributed––which was confirmed using histograms and Quantile-Quantile probability plots. Correlation tests were conducted using the cor.test function in R, and ggplot2 was used to cre-ate scatterplots.

To determine the ranks for urban centre green attributes, principal component analysis (PCA) was used to combine the three different measures of greenness (NDVI, tree cover, and OS Greenspaces) into a single measure for comparison across urban centres. Analysis was undertaken within SPSS (version 26). To test the robustness of the PCA approach—compari-sons were made with a standard Z-score approach. To do this, we obtained the mean and stan-dard deviation values for each variable and used the mutate function in R to generate Z-scores. This was carried out using the tidyverse package dplyr (version 4.0.2; [46]). Ranks for each urban centre were generated using Z-scores for individual green attributes (tree cover, green-ness, OS Greenspaces), and summed ranks were generated to provide an index of overall scores. Spearman's Rank correlation was used, which identified a very strong association between the PCA single greenness measure and Z-score output (df = 64, $R_s$ = 0.99, $P$ = <0.01). The PCA was chosen as the preferred method as it accounts for the degree of interrelationship between variables (particularly evident between the NDVI and tree cover measures and which could lead to double counting certain components within the Z-score approach).

The following flowchart (Fig 2) summarises the workflow for the experimental design including data collection parameters, decision making and analysis.

## Results

### Overall urban centre ranking for green attributes

The PCA identified a single factor with an eigenvalue above 1.0, accounting for just over 70% of the variance in the three input greenness measures. The loadings were: Urban centre μ NDVI 0.94; Tree cover (%) 0.87; and OSGS (%) 0.68. This demonstrated a strong correlation between variables, particularly between the NDVI and tree cover measures. Based on the results of the PCA rankings for each of the green attributes (tree cover, greenness, OS Green-space), the urban centre of Exeter in Devon, England, was the greenest (out of 68), with a mean NDVI of 0.15 (ranked 1 overall), a tree coverage of 11.67% (ranked 2 overall), and an OS Greenspace coverage of 0.05% (ranked 3 overall) (Fig 3).

The values for each of the green attributes for each urban centre are listed below in Table 1. It is interesting to note that at least the top 5 ranked urban centres are all located in the south of England, and the bottom 5 ranked urban centres all relate to ex-industrial areas in the north of Great Britain.

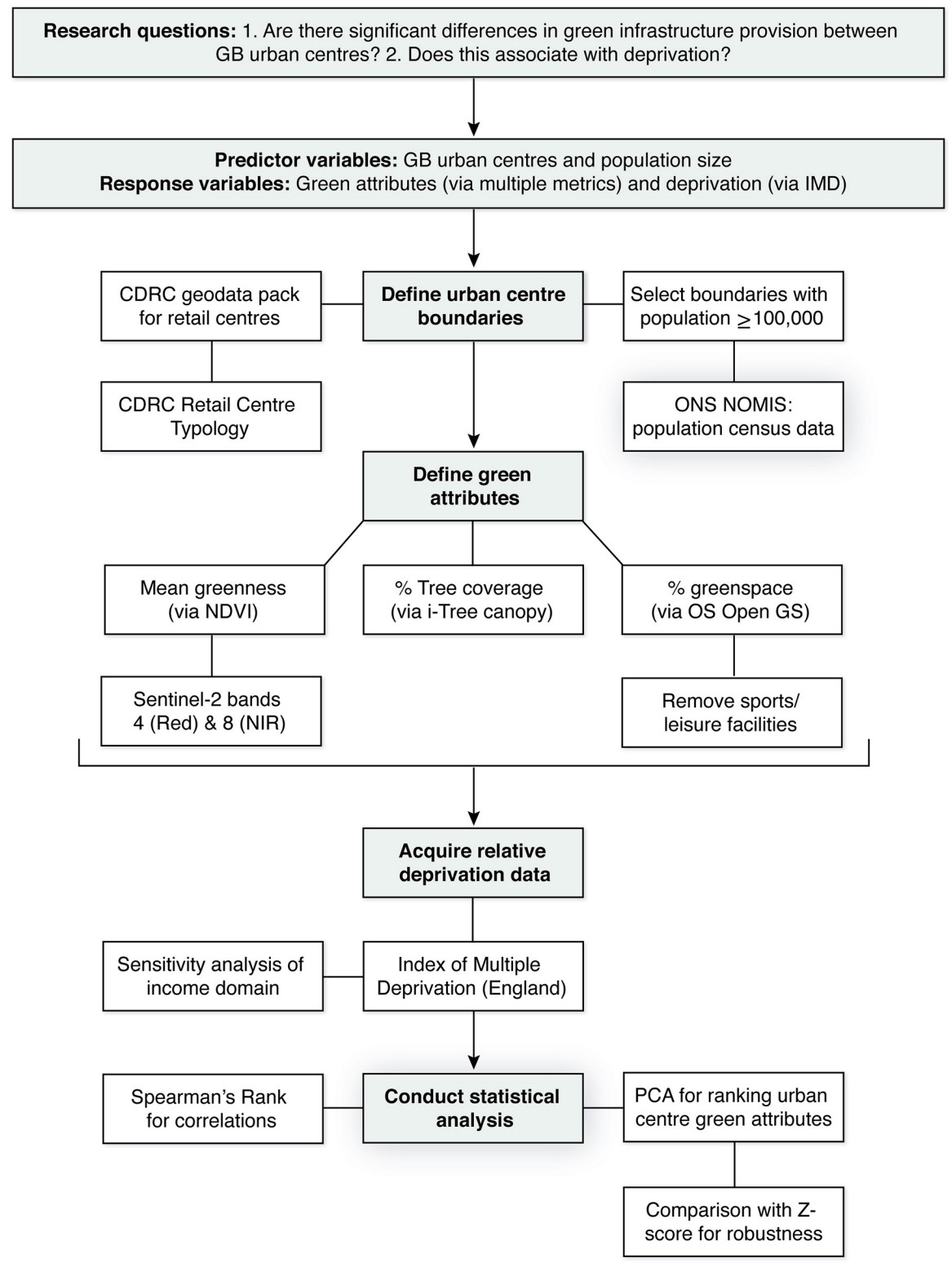

**Fig 2. Project workflow.**

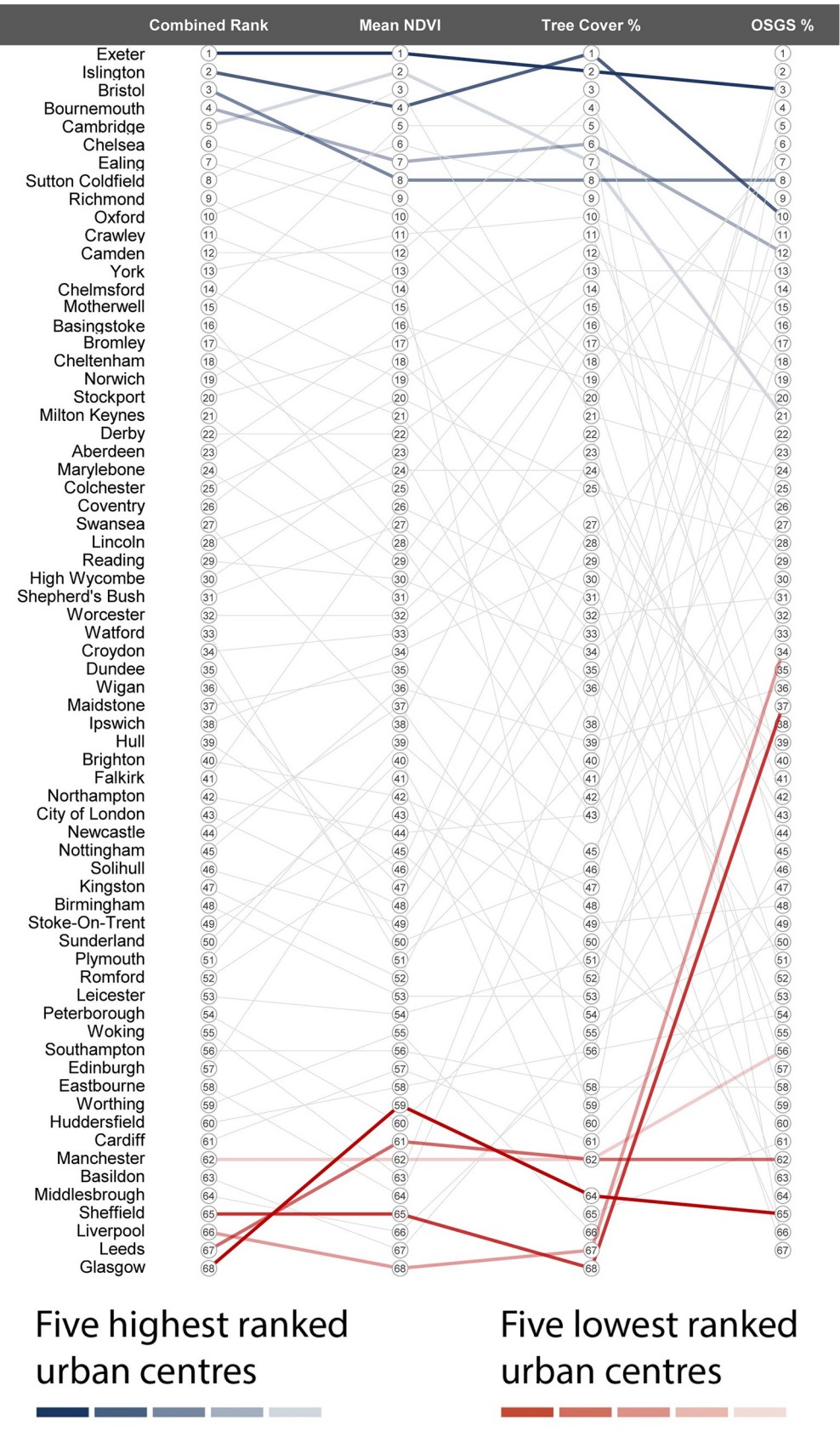

**Fig 3. Parallel coordinate plot showing all the selected GB urban centres in descending order of their combined green attribute ranking (based on the PCA)–with Exeter in the top position.** The chart also highlights the top 5 ranking urban centres (in blue), the bottom five ranking urban centres (in red), and the ranks for individual green attributes. OSGS = OS Greenspace.

## Comparing greenness (NDVI) in urban centres with the wider urban area (1 km radius)

As mentioned, one objective was to determine whether the level of greenness within urban centres is reflected across the wider urban area (1 km radius). Our results show a moderate positive correlation between the level of greenness within urban centres and that of the 1 km wider urban area (df = 64, $R_s$ = 0.49, $P$ = <0.01). Fig 4 highlights the five most green and five least green urban centres. Fig 5 shows a quadrat scatter plot, which indicates the distribution of the correlation data points. It also provides a visual output to comparatively assess the differences within and between urban centres. For example, Liverpool ranked relatively low on both NDVI values for the urban centre and the 1 km radius (μ NDVI = 0.01 and 0.12, respectively) thus Liverpool is in the bottom left quadrat. This quadrat indicates below mean NDVI values for both urban centre and 1 km radius. Whereas Sutton Coldfield ranked highly on both NDVI values for the urban centre and 1 km radius (μ NDVI = 0.15 and 0.69, respectively), thus, Sutton Coldfield falls in the top right quadrat. Woking ranked highly for NDVI values for a 1 km radius but low for an urban centre (μ NDVI = 0.02 and 0.60, respectively); thus, Woking is in the top left quadrat. Whereas Bristol ranked highly as an urban centre but relatively low for 1 km radius (μ NDVI = 0.13 and 0.30, respectively), thus, Bristol falls in the bottom right quadrat.

The mean urban centre NDVI was strongly associated with % tree coverage (df = 64, $R_s$ = 0.80, $P$ = <0.01) (S1 Fig).

## Urban centre size (population and area) and level of green attributes

Our results show a relationship between the urban centre population size and level of greenness and tree coverage, but not between population size and publicly accessible greenspaces. There was a moderate negative correlation between population size with three of our different greenspace measures: level of greenness within urban centres (df = 64, $R_s$ = -0.28, $P$ = 0.02); greenness of the wider urban area (1 km radius) (df = 64, $R_s$ = -0.39, $P$ = <0.01); and tree coverage (df = 64, $R_s$ = -0.34, $P$ = <0.01). However, there was no association between population size and area of OS Greenspace (df = 64, $R_s$ = -0.12, $P$ = 0.30) (Fig 6).

Our results reveal no relationship between the urban centre area and level of urban centre greenness (df = 64, $R_s$ = -0.20, $P$ = 0.10), 1 km greenness (df = 64, $R_s$ = -0.05, $P$ = 0.65), and tree coverage (df = 64, $R_s$ = -0.15, $P$ = 0.20), but do show a relationship between urban centre size and publicly accessible greenspaces. There was a moderate negative correlation between urban centre area ($m^2$) and area of OS Greenspace (df = 64, $R_s$ = -0.35, $P$ = <0.01) (Fig 7). The influence of a small number of data points on this relationship, however, should be noted.

## Relative deprivation and level of green attributes

For English urban centres, the correlation between the PCA greenness measure and index of multiple deprivation showed a weak to moderate negative relationship (df = 58, $R_s$ = -0.36, $P$ = <0.01)–whereby more deprived urban centres were found to be generally less green (Fig 8).

In order to explore any potential effects of intercorrelation (due to the IMD measure including measures related to accessibility), sensitivity analysis was undertaken using the

**Table 1. GB urban centres and their green attribute scores in alphabetical order.**

| Urban Centre | Urban centre μ NDVI | Urban centre NDVI (Z-score) | 1 km μ NDVI | 1 km NDVI (Z-score) | Tree cover (%) | Tree cover (Z-score) | OSGS (%) | OSGS (Z-score) | PCA greenness |
|---|---|---|---|---|---|---|---|---|---|
| Aberdeen | 0.09 | 0.66 | 0.33 | -0.18 | 8.48 | 1.87 | 0.01 | 2.68 | 0.45 |
| Basildon | 0.01 | -1.31 | 0.44 | 0.74 | 2.67 | 2.77 | 0.01 | 1.07 | -1.21 |
| Basingstoke | 0.07 | 0.10 | 0.49 | 1.16 | 4.32 | 1.38 | 0.06 | 1.33 | 0.91 |
| Birmingham | 0.03 | -0.83 | 0.21 | -1.19 | 2.88 | 1.45 | 0.02 | 0.89 | -0.71 |
| Bournemouth | 0.13 | 1.59 | 0.57 | 1.83 | 10.37 | 1.41 | 0.03 | 0.24 | 1.64 |
| Brighton | 0.05 | -0.50 | 0.22 | -1.11 | 3.91 | 0.92 | 0.02 | 1.51 | -0.35 |
| Bristol | 0.13 | 1.56 | 0.30 | -0.43 | 10.15 | 0.73 | 0.04 | 1.63 | 1.86 |
| Bromley | 0.09 | 0.50 | 0.54 | 1.58 | 5.53 | -0.25 | 0.04 | 2.70 | 0.80 |
| Cambridge | 0.15 | 2.22 | 0.50 | 1.24 | 10.25 | 1.74 | 0.02 | 0.30 | 1.61 |
| Camden | 0.11 | 1.09 | 0.34 | -0.10 | 11.49 | 1.19 | 0.01 | 0.46 | 1.07 |
| Cardiff | 0.02 | -1.06 | 0.17 | -1.53 | 2.05 | -0.46 | 0.01 | 3.28 | -1.18 |
| Chelmsford | 0.09 | 0.56 | 0.42 | 0.57 | 9.00 | 0.48 | 0.03 | 0.12 | 1.01 |
| Chelsea | 0.12 | 1.38 | 0.22 | -1.10 | 8.68 | 1.33 | 0.04 | -0.49 | 1.56 |
| Cheltenham | 0.08 | 0.30 | 0.48 | 1.10 | 9.07 | 1.81 | 0.02 | -0.44 | 0.66 |
| City of London | 0.04 | -0.72 | 0.16 | -1.61 | 6.06 | 1.02 | 0.01 | 0.82 | -0.43 |
| Colchester | 0.09 | 0.56 | 0.42 | 0.57 | 6.15 | -0.08 | 0.02 | 1.90 | 0.39 |
| Coventry | 0.09 | 0.57 | 0.30 | -0.43 | 6.05 | 1.52 | 0.02 | -1.32 | 0.38 |
| Crawley | 0.10 | 0.77 | 0.56 | 1.75 | 5.00 | -1.17 | 0.05 | 1.97 | 1.10 |
| Croydon | 0.06 | -0.25 | 0.38 | 0.23 | 6.15 | 1.04 | 0.01 | 0.30 | -0.19 |
| Derby | 0.09 | 0.50 | 0.31 | -0.35 | 8.97 | 1.06 | 0.01 | -0.25 | 0.51 |
| Dundee | 0.04 | -0.72 | 0.21 | -1.19 | 4.00 | 0.08 | 0.03 | 0.52 | -0.20 |
| Ealing | 0.12 | 1.28 | 0.44 | 0.74 | 8.09 | 0.99 | 0.04 | 0.08 | 1.48 |
| Eastbourne | 0.02 | -1.22 | 0.35 | -0.01 | 4.48 | 1.01 | 0.00 | -0.47 | -1.11 |
| Edinburgh | 0.05 | -0.49 | 0.33 | -0.18 | 1.94 | 0.94 | 0.00 | 0.29 | -1.10 |
| Exeter | 0.15 | 2.28 | 0.42 | 0.57 | 11.67 | 0.94 | 0.05 | -0.05 | 2.54 |
| Falkirk | 0.07 | 0.23 | 0.40 | 0.40 | 5.57 | 0.85 | 0.00 | -0.70 | -0.40 |
| Glasgow | 0.02 | -1.08 | 0.27 | -0.68 | 1.95 | 0.11 | 0.00 | 0.04 | -1.44 |
| High Wycombe | 0.08 | 0.38 | 0.50 | 1.24 | 4.44 | -0.20 | 0.02 | 0.46 | 0.05 |
| Huddersfield | 0.02 | -1.07 | 0.41 | 0.49 | 2.50 | 0.28 | 0.01 | -0.49 | -1.12 |
| Hull | 0.04 | -0.65 | 0.20 | -1.27 | 6.74 | -0.42 | 0.01 | 0.15 | -0.34 |
| Ipswich | 0.05 | -0.29 | 0.42 | 0.57 | 6.00 | -0.09 | 0.01 | 0.12 | -0.32 |
| Islington | 0.14 | 1.80 | 0.29 | -0.52 | 14.52 | -0.56 | 0.03 | 1.27 | 2.30 |
| Kingston | 0.03 | -0.81 | 0.33 | -0.18 | 5.79 | -0.41 | 0.01 | 0.91 | -0.57 |
| Leeds | 0.02 | -1.09 | 0.21 | -1.19 | 2.00 | 0.33 | 0.00 | -0.80 | -1.43 |
| Leicester | 0.03 | -0.84 | 0.24 | -0.94 | 3.16 | 0.06 | 0.01 | -0.40 | -0.92 |
| Lincoln | 0.08 | 0.37 | 0.32 | -0.26 | 6.67 | -0.59 | 0.01 | 0.30 | 0.10 |
| Liverpool | 0.01 | -1.47 | 0.12 | -1.95 | 1.06 | 0.30 | 0.01 | -0.49 | -1.42 |
| Maidstone | 0.05 | -0.30 | 0.40 | 0.40 | 4.24 | -0.06 | 0.02 | -1.02 | -0.30 |
| Manchester | 0.02 | -1.14 | 0.19 | -1.36 | 2.00 | -0.49 | 0.01 | -0.11 | -1.18 |
| Marylebone | 0.06 | -0.03 | 0.28 | -0.60 | 8.76 | -0.07 | 0.02 | -0.16 | 0.39 |
| Middlesbrough | 0.01 | -1.30 | 0.21 | -1.19 | 2.50 | -0.46 | 0.01 | 0.04 | -1.23 |
| Milton Keynes | 0.07 | 0.03 | 0.51 | 1.33 | 8.92 | -0.32 | 0.02 | -0.29 | 0.53 |
| Motherwell | 0.13 | 1.75 | 0.38 | 0.23 | 10.58 | 0.11 | 0.00 | -0.92 | 0.93 |
| Newcastle | 0.05 | -0.36 | 0.37 | 0.15 | 3.06 | 0.08 | 0.02 | -0.44 | -0.46 |
| Northampton | 0.04 | -0.54 | 0.30 | -0.43 | 4.32 | -0.86 | 0.02 | -0.18 | -0.41 |
| Norwich | 0.11 | 1.09 | 0.38 | 0.23 | 2.10 | -0.10 | 0.04 | -0.64 | 0.58 |

*(Continued)*

**Table 1.** (Continued)

| Urban Centre | Urban centre μ NDVI | Urban centre NDVI (Z-score) | 1 km μ NDVI | 1 km NDVI (Z-score) | Tree cover (%) | Tree cover (Z-score) | OSGS (%) | OSGS (Z-score) | PCA greenness |
|---|---|---|---|---|---|---|---|---|---|
| Nottingham | 0.05 | -0.36 | 0.37 | 0.15 | 4.75 | -0.92 | 0.01 | 0.13 | -0.48 |
| Oxford | 0.13 | 1.64 | 0.41 | 0.49 | 10.00 | 0.00 | 0.01 | -0.80 | 1.10 |
| Peterborough | 0.02 | -1.09 | 0.35 | -0.01 | 3.51 | -0.68 | 0.01 | -0.58 | -0.99 |
| Plymouth | 0.05 | -0.38 | 0.25 | -0.85 | 3.83 | -0.62 | 0.00 | -1.02 | -0.85 |
| Reading | 0.07 | -0.01 | 0.33 | -0.18 | 5.50 | -0.83 | 0.02 | -0.39 | 0.08 |
| Richmond | 0.10 | 0.88 | 0.43 | 0.65 | 11.29 | -0.95 | 0.02 | -0.63 | 1.18 |
| Romford | 0.04 | -0.55 | 0.37 | 0.15 | 2.78 | -1.13 | 0.01 | -0.74 | -0.86 |
| Sheffield | 0.02 | -1.28 | 0.33 | -0.18 | 0.33 | -1.04 | 0.01 | -0.28 | -1.40 |
| Shepherd's bush | 0.07 | 0.14 | 0.26 | -0.77 | 5.14 | -1.30 | 0.02 | -0.49 | 0.03 |
| Solihull | 0.04 | -0.72 | 0.63 | 2.34 | 5.45 | -1.22 | 0.01 | -0.95 | -0.51 |
| Southampton | 0.03 | -0.95 | 0.18 | -1.44 | 2.22 | -0.20 | 0.01 | -1.32 | -1.04 |
| Stockport | 0.09 | 0.59 | 0.31 | -0.35 | 9.14 | -0.72 | 0.01 | -0.95 | 0.53 |
| Stoke-on-Trent | 0.04 | -0.53 | 0.32 | -0.26 | 3.62 | -0.98 | 0.01 | -0.51 | -0.75 |
| Sunderland | 0.05 | -0.38 | 0.25 | -0.86 | 2.21 | -1.20 | 0.01 | -0.61 | -0.82 |
| Sutton Coldfield | 0.15 | 2.18 | 0.69 | 2.85 | 7.30 | -1.10 | 0.02 | -0.83 | 1.22 |
| Swansea | 0.05 | -0.36 | 0.29 | -0.52 | 8.75 | -0.41 | 0.02 | -.132 | 0.28 |
| Watford | 0.04 | -0.75 | 0.37 | 0.15 | 4.50 | -1.04 | 0.03 | -0.77 | -0.13 |
| Wigan | 0.04 | -0.70 | 0.31 | -0.35 | 5.56 | -1.20 | 0.02 | -0.80 | -0.24 |
| Woking | 0.02 | -1.16 | 0.60 | 2.08 | 5.14 | -1.50 | 0.00 | -0.27 | -1.02 |
| Worcester | 0.06 | -0.06 | 0.41 | 0.49 | 6.84 | -1.20 | 0.01 | -0.97 | -0.10 |
| Worthing | 0.03 | -0.84 | 0.27 | -0.68 | 1.67 | -1.73 | 0.01 | -0.31 | -1.11 |
| York | 0.11 | 1.26 | 0.35 | -0.01 | 9.56 | -1.22 | 0.02 | -1.25 | 1.06 |

Income domain component of the IMD. This analysis supported the previous findings, identifying a similar correlation between PCA greenness and deprivation (df = 58, $Rs$ = -0.32, $P$ = <0.01).

## Discussion

In this study, we conducted the first comparative assessment of the green attributes of GB urban centres. This is important because most research in this area has focused on suburban green infrastructure. Understanding potential disparities in green infrastructure provision in urban centres is vital to producing strategies that promote socio-ecological equity.

We ranked urban centres in GB based on their level of greenness, tree coverage and publicly accessible greenspaces. Our results highlight significant disparities in urban centre green attributes across GB. We reveal a significant positive association between urban centre greenness and greenness of the wider urban area (1 km radius) and a significant negative association between population size and urban greenness and tree coverage. We also found a significant weak to moderate negative association between IMD scores (a measure of deprivation) and overall greenness. A deeper exploration of these trends in a socioeconomic, health, and biodiversity context is warranted, as disparities in urban semi-natural environments play an important role in ecological justice––the equal and fair distribution of environmental resources and benefits.

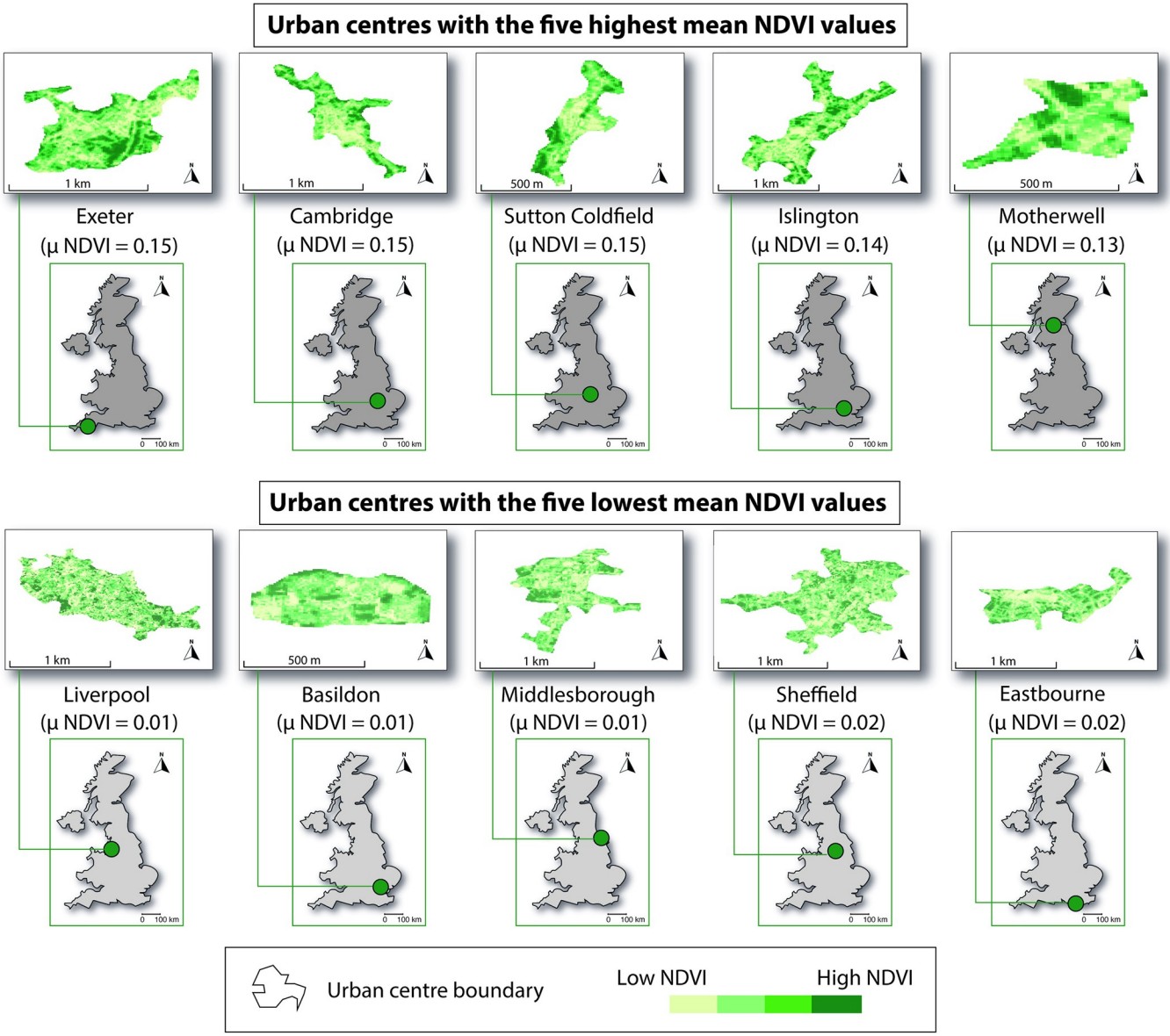

**Fig 4. Urban centres with the five highest and five lowest NDVI values.** The inset maps show the location of the corresponding urban centre in Great Britain. Map boundaries for the NDVI plots were generated in QGIS using the CDRC national-level geodata packs [32]. The figure is licensed under a CC BY 4.0 License.

### Urban centre green attribute ranking

The green attribute ranking process provides important baseline information. These data can help relevant stakeholders to monitor greening interventions in GB urban centres. They may also provide an incentive (particularly to the lower-ranked urban centres) to develop such interventions. Additionally, this process highlighted potential disparities in the presence/abundance of green attributes in urban centres across GB. This has important socioecological equity and justice implications, as green infrastructure is essential to human health and wellbeing. For example, spending time engaging with urban biodiversity is linked to reductions in stress and anxiety [39, 47, 48], improvements in positive affect [13], and immune regulation via

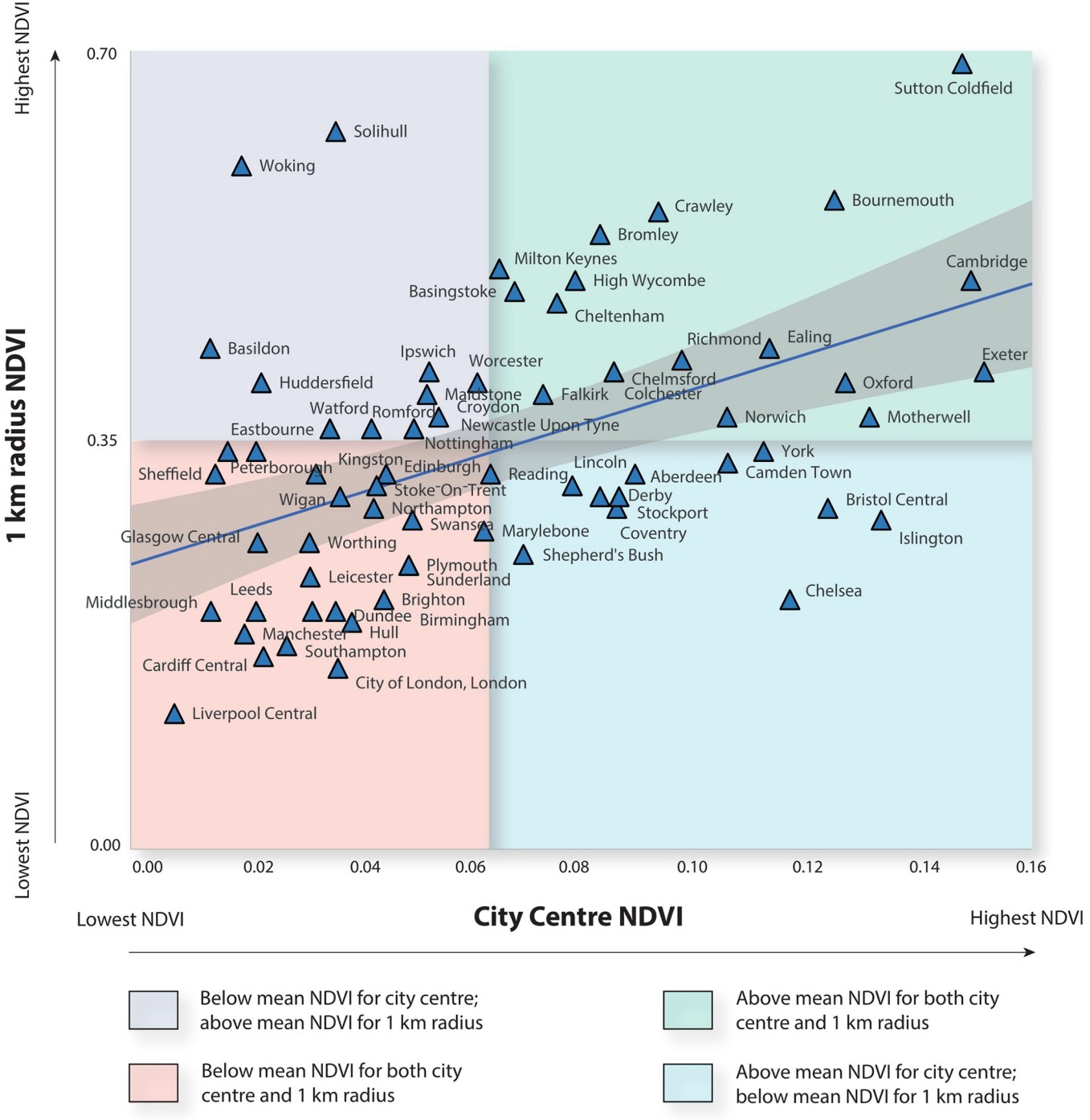

**Fig 5. Quadrat scatterplot showing the correlation datapoints, highlighting the within and between urban centre differences in mean NDVI values for both urban centres and the 1 km radii.**

microbial exposure [14]. Moreover, green infrastructure provides vital ecosystem services such as stormwater attenuation [49], urban cooling and climate change mitigation [9, 50], and buffering against pollution [51]. Additionally, disparities in these semi-natural habitats have important implications for biodiversity conservation efforts. We live in an epoch characterised

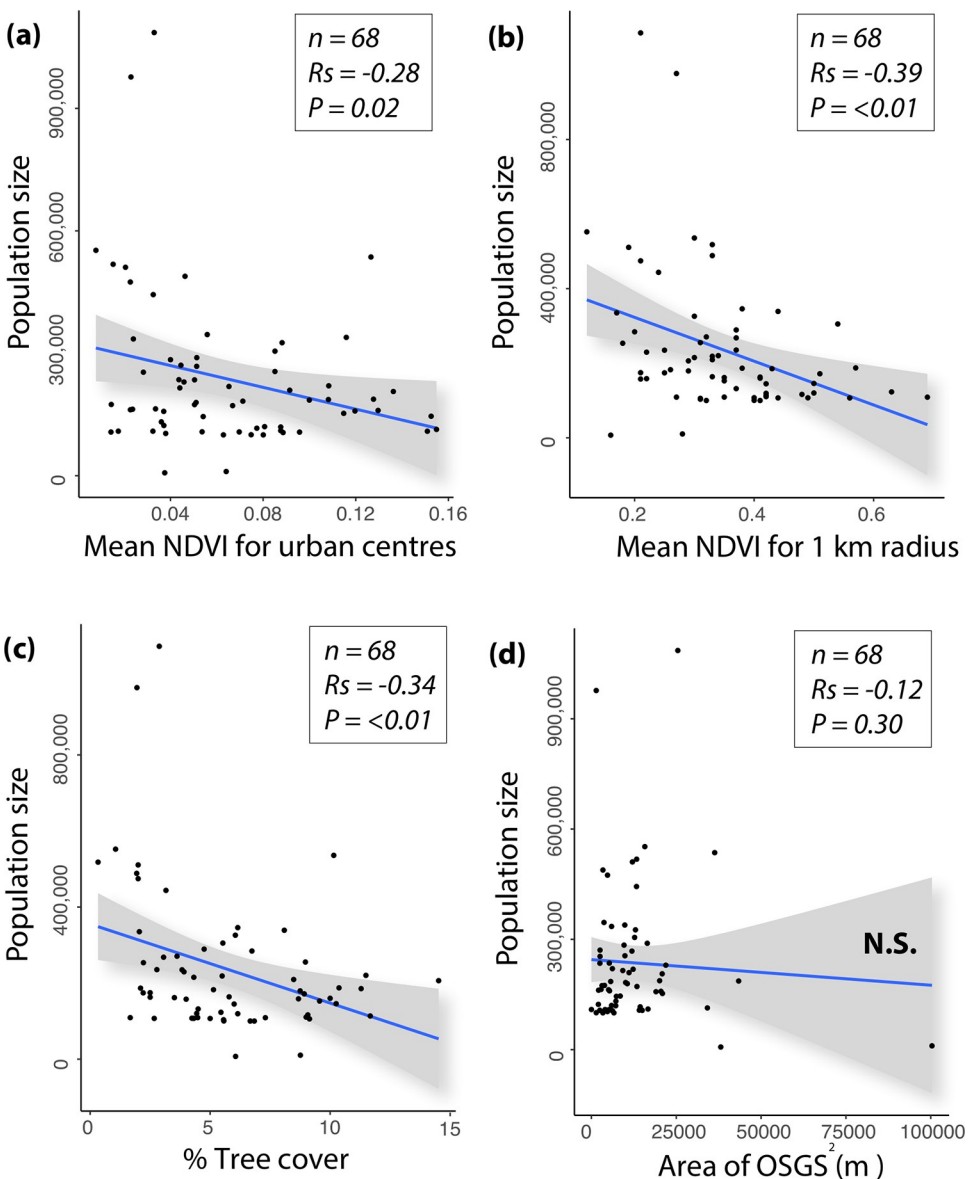

**Fig 6.** Correlation scatter plots for (a) population size and mean NDVI for urban centres; (b) population size and mean NDVI for 1 km radius; (c) population size and tree coverage; and (d) population size and area of OS Greenspace (OSGS). N.S. = not significant.

by a biodiversity crisis, partly due to habitat loss, landscape fragmentation [52, 53], and other factors associated with urbanisation such as air and light pollution [54]. Therefore, biodiversity needs enhanced and contiguous ecological, infrastructural, and societal support across the landscape including in urban centres, which can be neglected as our study shows.

Although not formally part of the analysis in this study, it is interesting to note that at least the five highest-ranked urban centres (for combined green attributes) are situated in the south of England, and the five lowest-ranked urban centres are in the north of England. Although further research is needed, other reports have demonstrated a north-south divide in terms of the abundance of trees in the wider landscape [55] along with significant socioeconomic and health status disparities. For example, Buchan et al. (2017) examined data on all deaths in

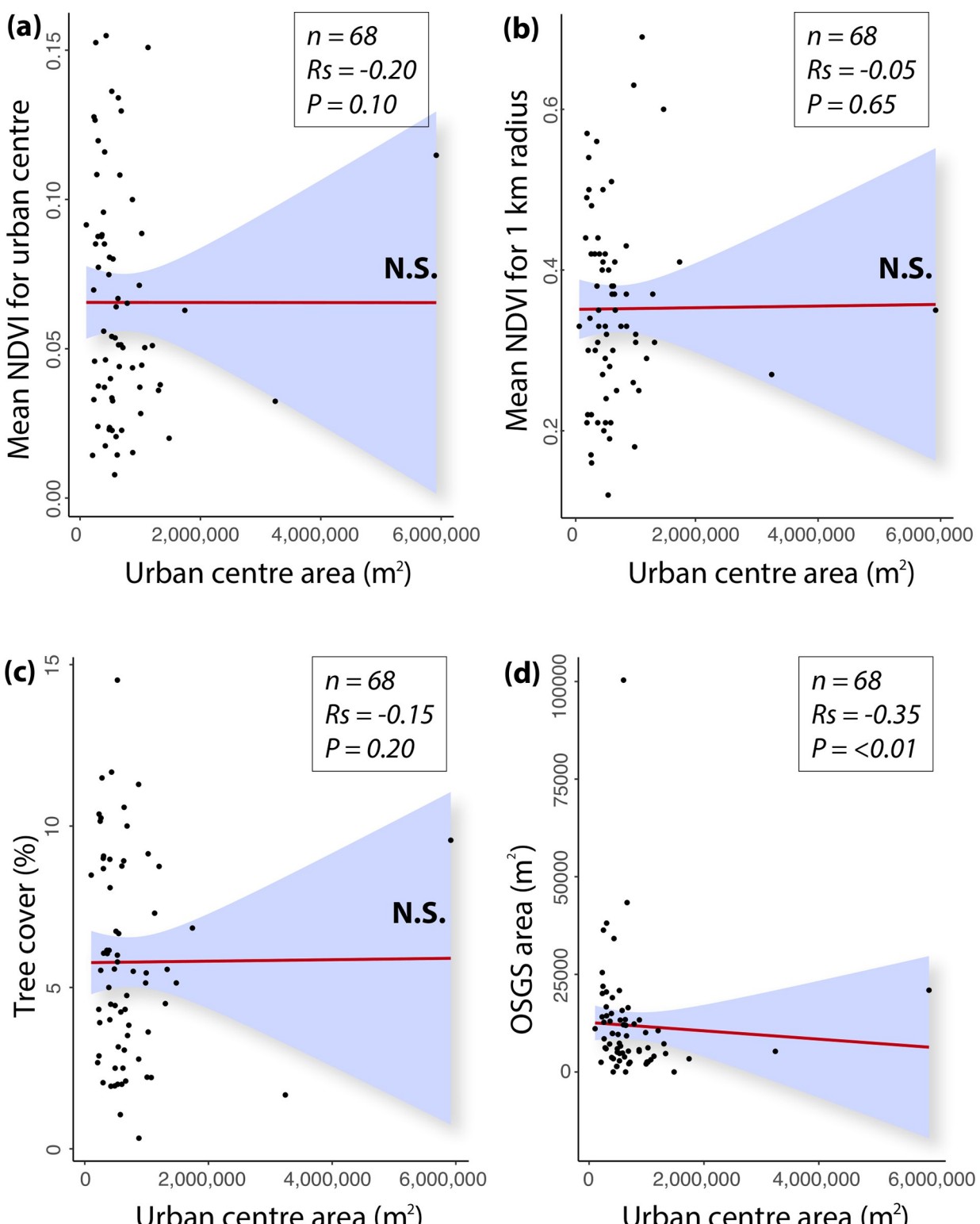

**Fig 7.** Correlation scatter plots for (a) urban centre area (m²) and mean NDVI for urban centres; (b) urban centre area and mean NDVI for 1 km radius; (c) urban centre area and tree coverage, and (d) urban centre area and area of OS Greenspace (OSGS). N.S. = not significant.

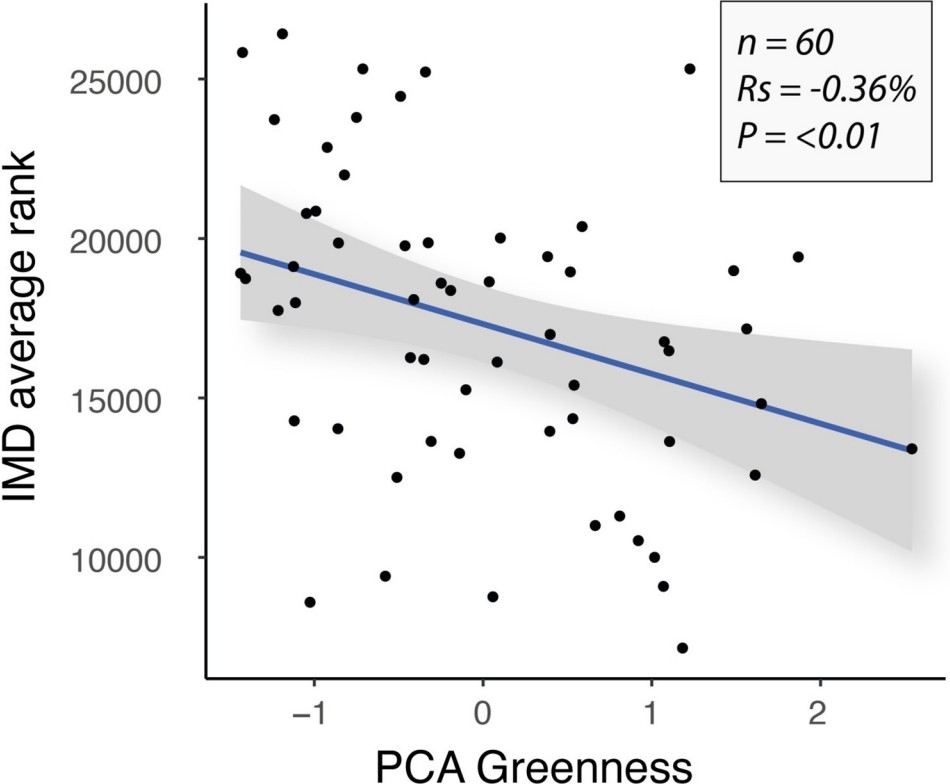

**Fig 8. Correlation scatter plot for IMD average rank and PCA greenness (combined green attribute scores) for English urban centres only.**

England between 1965–2015 [56] and discovered a 20% higher risk of dying aged <75 in the north of England. Given that green infrastructure is important for human health, this potential disparity is worth investigating further.

Urban centres in Great Britain are changing, and retail outlets are closing, mainly due to the evolution of digital shopping technologies. This has been accentuated by the COVID-19 pandemic, which has devasted the traditional retail property sector [57], resulting in the popularisation of the term 'end of the high street'. New parks, habitat corridors, nature-rich recreational facilities, and vertical farms––which bring immense value to humans, wildlife, and climate change mitigation––could potentially replace certain disused retail properties and vacant lots. As Albert Einstein purportedly said, "*in the middle of difficulty lies opportunity*" [58]. The high street crisis certainly presents a difficulty. Re-envisioning and re-developing urban centres to include enhancements in green infrastructure and biodiversity presents a potentially important opportunity. Indeed, although this is pertinent across the board, our study reveals that some urban centres are significantly lacking in health-promoting and biodiversity-supporting green attributes compared to others. Therefore, from an urban centre perspective, we provide an important indication of where greening support is most needed in GB. This information can potentially be used by the UK government and/or city-level authorities to reduce socio-ecological inequity. For instance, Members of Parliament, urban planners, and campaigners in the lower-ranked urban areas can use our study as an impetus to improve the quality of urban centres in these areas, particularly in the light of the levelling up agenda of the current UK government. The study can also be used as a platform by international researchers to explore potential disparities in urban centre green attributes in other countries.

## Urban centre and 1 km radius greenness (based on NDVI analysis)

We compared the results in the urban centres with 1 km radius to see if the greenness values were representative of the wider urban area. This provides additional valuable information on the distribution and potential disparities in green infrastructure provision across urban areas. For instance, it could indicate where planners/authorities should focus their greening efforts, and potentially central authorities could learn from the broader regions if disparities are recorded. It may also further emphasise the need to include urban centres in the bigger picture–as many studies focus on suburban areas (where people live) as opposed to the core (where many people spend much of their daily lives).

Our results show a moderate positive correlation between the level of greenness within urban centres and that of the wider urban area (1 km radius). This supports the hypothesis that greener urban centres, more broadly, may invest comparably more in the green attributes of their urban centres, whilst less green urban centres, more broadly, may invest relatively less. Yet it is likely that multiple drivers are leading to different levels of green attributes. Urban planning policy/strategy, policy implementation, supply-side constraints, political leadership, and other socioecological factors are likely important determinants of urban green infrastructure [59–61]. Future work should identify the multiple factors that impact urban centre greenness. This will provide foundational context to help understand how to improve and sustain the green attributes of lower-ranked urban centres. It could also be valuable to draw upon historical data to explore why some urban centres invested in parks and tree-lined avenues in the past. For example, in 19th century Britain, city planners often incorporated street trees. These decisions were influenced by the admiration of continental European boulevards and recognising the well-being benefits of 'garden cities' and 'spa towns' [62]. However, industry and war efforts contributed to urban sprawl and reduced natural features in certain urban centres, particularly in the North of Britain. Understanding both the historical and cultural context and community needs in the past and present, will likely be important to the success of future greening strategies. As Rotherham (2018, p.193) said:

> "*To really appreciate the importance of these trees [and other green attributes] and to understand how they should be managed, we need to recognise their historical and cultural significance*" [63].

With specific reference to NDVI greenness as a proxy for vegetation cover, the top 5 most green urban centres in this study were Exeter (Devon, England), Cambridge (Cambridgeshire, England), Sutton Coldfield (Birmingham, England), Islington (London, England), and Motherwell (Lanarkshire, Scotland). With the exception of Motherwell, these urban centres are considered to be relatively affluent [64], although deprivation may vary across wider geographic boundaries. Research has shown that the quality of greenspaces is higher in less socioeconomically deprived areas [18], which is consistent with these results. The urban centres with the lowest NDVI greenness in this study were Liverpool (Merseyside, England), Basildon (Essex, England), Middlesbrough (North Yorkshire, England), Sheffield (South Yorkshire, England), and Eastbourne (East Sussex, England). These urban centres have moderate to high levels of deprivation (although intra-urban centre variation occurs) [65, 66], which is also consistent with the above socioeconomic hypothesis.

Providing equitable access to health-promoting, biodiverse green infrastructure is vital to ensure we have flourishing, resilient communities. Populations from otherwise socioeconomically disparate areas spend considerable time in urban centres gathering and mingling for work, shopping, and recreation. Therefore, enhanced greening interventions in urban centres

may also reduce the inequality of opportunity for diverse populations in terms of nature-based, health-promoting pathways and impacts.

On a final note, perhaps expectedly, the mean urban centre NDVI scores strongly (and positively) associated with % tree coverage (S1 Fig). This suggests that trees are the predominant green features in urban centres in Great Britain. Future research could replicate the work, with additional analysis, for example, by using NDVI thresholds to identify green areas rather than applying mean NDVI values.

## Urban size (population and area) and green attributes

Our results show a moderate negative correlation between population size and level of greenness within urban centres and between population size and greenness of the wider urban area (1 km radius). This finding is potentially important because it indicates that per capita, health-promoting (and biodiversity-supporting) green attributes may reduce as population increases, thereby highlighting another socioecological justice issue. By 2050, it is expected that 70% of the world's population will be urbanised [67]. Indeed, 84% of the UK's (GB including Northern Ireland) population already lives in towns and cities [68], and the population size of these urban centres increases year upon year, with a current 2015–2025 growth rate projection of 7.6% [69]. It will be important for local authorities and urban planners to ensure the levels of urban centre green attributes do not decrease or remain inert as population size increases because they are important for human health and wellbeing. Furthermore, as population size increases, the pressures on biodiversity are also likely to increase due to expanding grey space and anthropogenic stressors (e.g. pollution) [70]. Therefore, it will be imperative to ensure urban centre green attributes play their role in sustaining biodiversity and enhancing habitat corridors across the city and into rural areas. There was also a moderate negative correlation between population size and tree coverage but no association between population size and area of OS Greenspace.

Regarding urban centre area (m$^2$) and green attributes, our results revealed no relationship between the size of urban centres and the extent of urban centre greenness, 1 km greenness, and tree coverage. However, they show a relationship between urban centre size and publicly accessible greenspaces in that urban centre area was negatively associated with the proportion of greenspace. This relationship (Fig 6D) is heavily influenced by a small number of data points and requires further work to support these findings. The lack of a correlation between urban centre size and greenness/tree cover is another potentially significant finding in the context of disparities between urban centres. For example, one may expect green attributes to increase proportionally to the size of the urban centre. In contrast, the non-correlative pattern observed in our results shows that many smaller urban centres had a relatively high level of green attributes, and many had a relatively low level of green attributes. Moreover, many larger urban centres had a relatively high level of green attributes, and many had a relatively low level of green attributes (as indicated in Fig 6). This result suggests inter-urban centre disparities in the level of green attributes (not based on size per se), which could again indicate socioecological injustice on a GB-wide scale.

## Relative deprivation

Our results show a weak to moderate significant negative association between deprivation and the overall greenness of urban centres. Whilst not a universal rule and requiring further research to confirm the relationship, generally speaking, in this study, more deprived urban centres were more likely to be less green than less deprived counterparts. Given the known associations between health, wellbeing and greenspace [13, 14, 23, 39, 48, 71], this has

important implications for current government policy and the desire for levelling up existing social inequalities. This is especially pertinent because disparities in quality living environments are critical drivers of health inequities [72, 73]. For example, people living in areas of higher deprivation are more likely to be exposed to poor air quality [74, 75] and poor quality greenspaces [18]. Therefore, the health impacts of exposure to these poor environmental conditions–and lack of access to better quality conditions––are also unequally spread across the socioeconomic spectrum, representing a major socio-ecological justice issue. These disparities demonstrate that transdisciplinary solutions are needed to promote equitable access to healthy living environments (e.g. accessible, safe, biodiverse greenspaces with clean air), along with policy changes that enforce monitoring and regulation of environmental conditions.

## Limitations

The study has several limitations. For instance, the satellite data used within the study was a composite dataset provided through the EDINA Digimap Service. It is therefore possible that geographic disparities may be present, for example, different data timeframes. Such limitations are outside of the control of this exploratory study. Some data were not available, for instance, IMD for all countries and green metrics for Northern Ireland (hence being omitted). Other vegetation indices (such as the EVI) could also provide different results in urban areas and should be considered in future studies. The restricted scope of our study (e.g., focusing on GB and sampling urban areas with >100,000 population) means the results should be extrapolated with caution.

## Conclusion

This is the first known study to comparatively define and rank urban centres in Great Britain based on multiple green attributes. The results suggest significant differences in the proportion of green attributes between urban centres. The finding that population size is negatively associated with greenness and tree coverage within urban centres suggests a relative diminishment of health-promoting and biodiversity-supporting resources as population increases. Furthermore, urban centre greenness and relative deprivation were also negatively associated. These disparities in green infrastructure across the country, along with the population and deprivation-associated trends, are important in the realms of socioecological and equity justice. For example, the current non-communicable disease crisis and the biodiversity crisis highlight the need to ensure the presence of, and equitable access to, quality green spaces across our landscapes. Ecologically conscious greening interventions in urban centres could play a vital role in supporting both human health (and reducing inequality of opportunity by reaching diverse populations) and biodiversity. The need to re-imagine and re-develop our urban/city centres due to digital shopping technologies and societal changes provides an important opportunity to explicitly consider the enhancement of urban centre biodiversity. This study provides a baseline and stimulus to help local authorities and urban planners create and monitor greening interventions in urban centres.

## Supporting information

**S1 Fig. Scatterplot showing mean urban centre NDVI and tree cover (%) positive association.**
(TIF)

**S1 Table. List of urban centres and their greenspaces as defined by the OS Greenspace (OSGS) dataset.**
(PDF)

**S1 Data.**
(ZIP)

## Acknowledgments

For the purpose of open access, the authors have applied a Creative Commons Attribution (CC BY) licence\* to any Author Accepted Manuscript version arising. **Data access statement:** This study brought together existing research data obtained through a combination of Open Data (Index of Multiple Deprivation; Ordnance Survey Open Greenspace; Consumer Data Research Centre retail centre boundaries), data within Open-Source software (Landsat satellite imagery within i-Tree) and the Digimap educational data repository (Sentinal-2 satellite imagery). Digimap is an online service that provides maps and mapping data to UK colleges and universities and licence restrictions apply. The data have been deposited on Dryad: DOI https://doi.org/10.5061/dryad.p2ngf1vtj.

## Author Contributions

**Conceptualization:** Jake M. Robinson, Paul Brindley.

**Data curation:** Jake M. Robinson, Kate Robinson, Paul Brindley.

**Formal analysis:** Jake M. Robinson, Kate Robinson, Paul Brindley.

**Funding acquisition:** Paul Brindley.

**Investigation:** Jake M. Robinson, Paul Brindley.

**Methodology:** Jake M. Robinson, Paul Brindley.

**Project administration:** Jake M. Robinson.

**Resources:** Jake M. Robinson.

**Supervision:** Paul Brindley.

**Validation:** Jake M. Robinson.

**Visualization:** Jake M. Robinson, Kate Robinson, Paul Brindley.

**Writing – original draft:** Jake M. Robinson, Suzanne Mavoa, Paul Brindley.

**Writing – review & editing:** Jake M. Robinson, Suzanne Mavoa, Kate Robinson, Paul Brindley.

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
