## [Decision Letter · Decision Letter 0]

22 Jun 2022

PONE-D-21-34255Urban Centre Greenness, Tree Cover, and Green Spaces in Great Britain: A Geospatial StudyPLOS ONE

Dear Dr. Robinson,

Thank you for submitting your manuscript to PLOS ONE. After careful consideration, we feel that it has merit but does not fully meet PLOS ONE’s publication criteria as it currently stands. Therefore, we invite you to submit a revised version of the manuscript that addresses the points raised during the review process.

Specifically, there are mainly some major issues among which the reviewers point to the objectives of the study, its main contribution and other issues about methodology, results and conclusion. The details of the reviews can be seen in the reviewers' reports.

We look forward to receiving your revised manuscript.

Kind regards,

Eda Ustaoglu, PhD

Academic Editor

PLOS ONE

Journal Requirements:

“SM was supported by a University of Melbourne Faculty of Medicine, Dentistry and Health Sciences Research Fellowship.”

3. We note that Figure 3 in your submission contain [map/satellite] images which may be copyrighted. All PLOS content is published under the Creative Commons Attribution License (CC BY 4.0), which means that the manuscript, images, and Supporting Information files will be freely available online, and any third party is permitted to access, download, copy, distribute, and use these materials in any way, even commercially, with proper attribution. For these reasons, we cannot publish previously copyrighted maps or satellite images created using proprietary data, such as Google software (Google Maps, Street View, and Earth). For more information, see our copyright guidelines: http://journals.plos.org/plosone/s/licenses-and-copyright.

 a. You may seek permission from the original copyright holder of Figure(s) [#] to publish the content specifically under the CC BY 4.0 license. 

Reviewers' comments:

Reviewer's Responses to Questions

**Comments to the Author**

1. Is the manuscript technically sound, and do the data support the conclusions?

Reviewer #1: Partly

Reviewer #2: Partly

Reviewer #3: Partly

Reviewer #4: Yes

2. Has the statistical analysis been performed appropriately and rigorously? 

Reviewer #1: Yes

Reviewer #2: Yes

Reviewer #3: Yes

Reviewer #4: N/A

3. Have the authors made all data underlying the findings in their manuscript fully available?

Reviewer #1: Yes

Reviewer #2: Yes

Reviewer #3: Yes

Reviewer #4: Yes

4. Is the manuscript presented in an intelligible fashion and written in standard English?

Reviewer #1: Yes

Reviewer #2: Yes

Reviewer #3: Yes

Reviewer #4: Yes

5. Review Comments to the Author

Reviewer #1: Urban Centre Greenness, Tree Cover, and Green Spaces in Great Britain: A Geospatial Study

This paper uses remote sensing and land use data to assess greenspace quantity in UK. It is an interesting paper, but there are some problems with the empirical parts.

Introduction

1.The objective of this research is not clear enough. Do the authors want to focus on developing new green space metrics or disparities in green space provision？

2.The contribution is not also very vague. For example, what geospatial insights does this study provided regarding the green space provision in UK?

Methods

1.What are differences among these three green space metrics? Why did the authors use different metrics?

2.Which metric reflects the accessible greenspaces?

3.IMD is composite metric, which involves sub-domain of accessibility item. How did the authors eliminate the influence of such embedded issue of IMD?

4.In Scotland, Scottish Index of Multiple Deprivation is normally used to measure deprivation, which is different from IMD. How did the authors integrate these two index to make it comparable?

5.The correlation coefficient among different green space metrics should be displayed.

6.Why wasn’t the relative deprivation weighted by population, but weighted by area of the district?

Results

1.In Fig 3 and 4, why only the result of NDVI was presented? What about other two metrics?

2.Why did the authors compare the result in city centre and 1kn radius?

Discussion

Compared with existing literature, what are the new findings from this study should be highlighted.

Reviewer #2: 1. The MS does contribute new in terms of methodology – but a set of well-known methods are available to apply on the research area and their physical parameters. I fail to see any fruitful discussion on the generated datasets. The scientific problem is analyzed and solved. The introduction part is clear and the scientific problem has clearly identified and addressed.

2. The authors discuss clear point about the Greenness tree cover but not emphasize on application - they have studied Great Britain area and their MS used the Geospatial application.

3. The introduction is good, the method section is trivial and vague at places. Discussion is existing but not extensive. Increase the literature and cite in the introduction part (more than 70 papers cited which is more enough but I find many papers not related to the study work. Please remove extra cited papers and be relevant to the research topic).

4. I don't feel qualified to judge about the English language but the manuscript style needs improvement according to the authors guidelines.

5. Abstracts and Introduction does give a concise information about objective of this work. Why is it needed, what novel is in this research, which papers concerned these research questions to the greenery in the Great Britain, and why? The reasons to study the chosen area in well explained? Why greenery changes in Great Britain region are important (besides urbanization)? Which studies are for this region? Partially, these questions were answered, and in my opinion, it is enough and quite efficiently.

6. Data source and processing section is completed. Authors can add more information about the datasets acquired (if it feels necessary, software used, processing methods applied to the data: which one is the final resolution as the research was conducted with datasets at different spatial resolutions? A reference is needed for the different indices.

7. Description of methodology is concise. Maybe, the classification technique could be visualized or/and quantified based on more example?

Recommendation. Summarize my comments, Overall, the manuscript is written well. I would recommend to Authors to address my comments into manuscript taking into account most of issues including motivation, methods, results analysis. These are the major revisions and after the complete elaboration, the manuscript could be accepted in this Journal.

Reviewer #3: The Paper presents a comprehensive and detailed evaluation of urban centers' green infrastructure. The Paper has the potential to be published but some essential points that I listed below should be reconsidered or revised.

1. Abstract should support the findings with quantitative analysis results.

2. I believe remote sensing based green cover identification in city centers needs a short paragraph in the Introduction. There are several attempts on this topic, using different methodologies and different vegetation indices, and few of them should be cited.

3. Methods section clearly needs a flowchart including data, implemented methods for parameter extraction, main methodology for decision making, and result representations.

4. Authors should explain why they choose a specific indices or method among others in a convincing way (superiority over other ones?) for example why you selected NDVI among other vegetation indices or why you selected PCA?

5. In lines 126-133, I do not think you need to give the software and their processing steps but you should explain the algorithm.

6. In lines 139-143, again how the data is collected and processed by who is not a concern, but technical details of the satellites and image frame dates are more important. Different seasons will affect the determination of vegetation, thus frame dates should be as closer as much in a country level mosaic.

6. In line 140, not "Sentinal" but "Sentinel".

7. In line 144, what do you mean by isolation, you do not need to separate the bands while calculating NDVI.

8. In line 147, formulae should be given as NDVI= (Near Infrared- Red)/( Near Infrared +Red) no need for the light term, and NDVI should be on the left-hand side.

9. The result of the NDVI is not a direct greenness score. For example, high density vegetation cover with lower chlorophyll content (due to season or other conditions) will provide lower values. Please take advice from a remote sensing professional on this issue.

10. "The data was clipped according to urban centre boundaries" will be enough in line 159.

11. Lines 166-172 needs rewriting for clear explanation and again more focus on the algorithm.

12. I highly encourage not to use mean NDVI directly as it will be affected by the distribution of land use classes and may not reflect the greenness of a city center. It is well observed in Figure 3. that Exeter has quite more green areas with respect to total cover but takes the same value as Cambridge or Sutton coldfield. Also, Basildon does not have that less green cover to take 0.01. I recommend applying a threshold for green cover, calculate the area of green cover, and ratio it according to the total area of that city boundary.

13. Figure qualities are low in general.

14. In discussion, please avoid long writings that belongs to other studies but not matching your findings directly (eg, lines 329-338).

15. Lastly and importantly, I would like to see maps of your findings. GIS tools can help you easily to perform these tasks.

Reviewer #4: The authors have applied geospatial techniques to compare green infrastructure among urban centers in Great Britain. This study is conducted with the aim that it helps stakeholders to make decisions regarding green infrastructure. The abstract is informative and the reference list captures all the citations in the text. However, a few corrections are needed;

1. The authors should proofread the entire manuscript to check for grammatical and typo errors. For instance “Sentinal-2” on line 140 should be “Sentinel-2”. On line 93, the sentence should read (Northern Ireland was excluded due to data unavailability).

2. In Figure 3, the authors omitted a scale and north arrow.

6. PLOS authors have the option to publish the peer review history of their article (what does this mean?). If published, this will include your full peer review and any attached files.

Reviewer #1: No

Reviewer #2: No

Reviewer #3: No

Reviewer #4: No

---

## [Author Response · Author response to Decision Letter 0]

1 Sep 2022

Submitted to PLOS ONE

Urban centre green metrics in Great Britain: A geospatial and socioecological study

Jake M. Robinson*, Suzanne Mavoa, Kate Robinson, and Paul Brindley

Editor/reviewer feedback and author responses.

Date: 20-07-2022

Comment

 Response

Editor

 Thanks again for considering our manuscript for PLOS ONE. We are pleased that you see value in our manuscript.

My co-authors and I have worked diligently to address each concern raised in the reviews (as you’ll see from the extensive responses below). In particular, we have revised and added additional information to the Abstract, Introduction, Methods and Discussion. This includes greater clarity regarding the aims and methodological approaches taken, a new workflow diagram, revisions to what was Fig. 3 (now Fig. 4) (to address any potential copyright issues with Google), further discussion points and stronger justification, along with greater emphasis on the novelty and importance of the study. 

We all agree it has greatly improved the manuscript and we hope you now find it in an acceptable form for publication in PLOS ONE.

Reviewer 1

This paper uses remote sensing and land use data to assess greenspace quantity in UK. It is an interesting paper, but there are some problems with the empirical parts.

 Many thanks for your thorough review of our manuscript. 

My co-authors and I have worked diligently to address each concern raised in your review, and we think your feedback has greatly improved the manuscript. 

Introduction

1.The objective of this research is not clear enough. Do the authors want to focus on developing new green space metrics or disparities in green space provision？

Thanks for this feedback. We have now ensured the objectives are clearer in the Abstract and Introduction sections. For instance, in the Abstract, we have reworked Lines 25-36, adding further context and clarity. We have also included additional text in the Introduction at Lines 113-120 to strengthen the objectives of investigating potential disparities in green infrastructure provision in GB. 

i.e., “However, little is known about the equity of green infrastructure provision in city centres. Disparities in green infrastructure could underscore an important socio-ecological justice issue, with some populations gaining the benefits of healthy urban ecosystems and others enduring the disbenefits of poor green infrastructure provision. The same applies to biodiversity––i.e., it is important to understand potential disparities in wildlife-supporting green infrastructure in city centres. Considering these factors, we argue that more emphasis should be placed on mapping and enhancing the green attributes (e.g., trees, broader vegetation cover, and publicly accessible greenspaces such as parks) of urban cores/centres. In doing so, there is considerable potential to provide a range of positive health benefits to many people across the socioeconomic spectrum via the augmented provision of health-promoting green features such as urban forests and recreational greenspaces. There is also potential to enhance biodiversity and interspecies health.”

We used different established metrics to study these potential disparities and combine them for robustness. The main aim, however, is to map and study potential disparities across city centres in GB (studies to date mostly focus on residential areas e.g., the suburbs where people live and often only use 1-2 metrics).

2.The contribution is not also very vague. For example, what geospatial insights does this study provided regarding the green space provision in UK? Many thanks for pointing this out. We have now ensured the manuscript emphasises the contribution to the literature. This is highlighted in discussions of the objectives as per above quotation, in the revised objectives section of the Introduction (Lines 141-156), e.g., “These objectives allowed us to map and understand potential disparities in green infrastructure provision in Great British urban centres.” To our knowledge, this has never been done before, especially using multiple metrics and specifically in urban centres.

Methods

1.What are differences among these three green space metrics? Why did the authors use different metrics?

 Thanks. The differences are explained in the Objectives section (Lines 141-156) and in the Methods section (Lines 166-270). 

The three metrics were used for robustness (Objective section; Line 144). The three metrics were based on open source data and were used to provide a comprehensive assessment of the green attributes in urban centres. 

We have also provided an additional figure with the methodological workflow for greater clarity (now Figure 2). 

2.Which metric reflects the accessible greenspaces? Thanks for this query. As now highlighted in the Greenspace attributes section (Lines 251-270), the Ordnance Survey (OS) Open Greenspace dataset contains the accessible greenspaces. 

i.e., “contains data for publicly accessible urban greenspaces in GB. Before analysis, we removed all greenspaces classified as sports/leisure facilities due to the high level of grey space (e.g. buildings such as leisure centres and carparks) and a low level of greenspace. See Appendix II for a full breakdown of OS Greenspaces in each urban centre.”

3.IMD is composite metric, which involves sub-domain of accessibility item. How did the authors eliminate the influence of such embedded issue of IMD? Thank-you for highlighting this. To to address your concern we have undertaken further analysis using just the income domain (a technique commonly employed within health analysis to counter potential intercorrelation with the health domain within the IMD). Results are consistent. We have clarified this point by adding the following sentence:

“In order to explore any potential effects of intercorrelation (due to the IMD measure including measures related to accessibility), sensitivity analysis was undertaken using the Income domain component of the IMD. This analysis supported the previous findings, identifying a similar correlation between PCA greenness and deprivation (df = 58, Rs = -0.32, P = <0.01).”

4.In Scotland, Scottish Index of Multiple Deprivation is normally used to measure deprivation, which is different from IMD. How did the authors integrate these two index to make it comparable? Thanks. We used the IMD for England only (n=60). We highlighted this in the Relative deprivation section (Lines 272-280). i.e., “Deprivation could only be tied to the 60 English urban centres as different incompatible data exist for the other countries of Great Britain.”

5.The correlation coefficient among different green space metrics should be displayed. The PCA loadings were for the green attribute metrics are stated in the Results section (Line 316-317) “Urban centre μ NDVI 0.94; Tree cover (%) 0.87; and OSGS (%) 0.68.”, and the correlation plot with the mean urban centre NDVI values and tree cover % correlation coefficient is presented in Appendix I – showing a strong significant (positive) correlation, indicating the greenness value mostly derives from the tree coverage in the urban centres. 

6.Why wasn’t the relative deprivation weighted by population, but weighted by area of the district? We decided to weight relative deprivation by area because the focus of this work rests not on residential locations that people live (relevant locations for population) but those in which people tend to concentrate daily activities within urban cities. These include locations of employment, recreation and leisure and so forth. As such, city/urban centres contain a wide diversity of land uses including both populated and non-populated areas. Any approach should not privilege populated areas at the expense of non-populated ones. Most centres fell within a single local district and therefore did not require weighting. In only three cases, weighting was necessary.

We have clarified this point by adding the following sentence at Line 295: “Three urban centres fell across multiple local authority districts. In these cases, weighting of deprivation was undertaken based on the area of the district. Area as opposed to population was used for the weight, as the focus of this work - urban centres, are inherently a blend of both populated and non-populated geographic areas.”

Results

1.In Fig 3 and 4, why only the result of NDVI was presented? What about other two metrics? Fig 3 and 4 in the Results are presented in the section that specifically covers greenness (via the NDVI metric) in urban centres and the wider urban area (1 km radius) for comparison. Therefore, only figures that show NDVI results are included in this section.

2.Why did the authors compare the result in city centre and 1km radius? We compared the results in the urban centres with 1 km radius to see if the greenness values were representative of the wider urban area. This provides additional valuable information on the distribution and potential disparities in green infrastructure provision across urban areas. For instance, it could indicate where planners/authorities should focus their greening efforts, and potentially central authorities could learn from the broader regions if disparities are recorded. It may also further emphasise the need to include urban centres in the bigger picture – as many studies focus on suburban areas (where people live) as opposed to the core (where many people spend much of the daily lives). This point is now included in the Discussion. 

Discussion

Compared with existing literature, what are the new findings from this study should be highlighted. Many thanks for your comments. We have now included additional discussion points on what our findings show. 

We have included additional text at Lines 426-429: “This is important because most research in this area has focused on suburban green infrastructure. Understanding potential disparities in green infrastructure provision in urban centres is vital to producing strategies that promote socio-ecological equity.”

Lines 462-467: “Additionally, disparities in green infrastructure have important implications for biodiversity conservation efforts. We live in an epoch characterised by a biodiversity crisis, partly due to habitat loss, landscape fragmentation [49,50], and other factors associated with urbanisation such as air and light pollution [51]. Therefore, biodiversity needs enhanced and contiguous ecological, infrastructural, and societal support across the landscape including in urban centres, which can be neglected as our study shows.”

Lines 499-508: “Indeed, although this is pertinent across the board, our study reveals that some urban centres are significantly lacking in health-promoting and biodiversity-supporting green attributes compared to others. Therefore, from an urban centre perspective, we provide an important indication of where green infrastructure support is most needed in GB. This information can potentially be used by the UK government and/or city-level authorities to reduce socio-ecological inequity. For instance, Members of Parliament, urban planners, and campaigners in the lower-ranked urban areas can use our study as an impetus to improve the quality of urban centres in these areas, particularly in the light of the levelling up agenda of the current UK government. The study can also be used as a platform by international researchers to explore potential disparities in urban centre green attributes in other countries.”

We have also included a limitations section at Lines 644-657:

“Limitations

It should be noted that the study has several limitations. For instance, the satellite data used within the study was a composite dataset (from 2019) provided through the EDINA Digimap Service. It is therefore possible that geographic disparities may be present, for example, different data timeframes. Such limitations are outside the control of this exploratory study. Some data were not available, for instance, IMD for all countries and green metrics for Northern Ireland (hence being omitted). Other vegetation indices (such as the EVI) could also provide different results in urban areas and should be considered in future studies. The restricted scope of our study (e.g., focusing on GB and sampling urban areas with >100,000 population) means the results should be extrapolated with caution.”

Thanks again for your valuable feedback. We all agree it has greatly improved the manuscript and we hope you now find it in an acceptable form for publication in PLOS ONE. 

Reviewer 2 

1. The MS does contribute new in terms of methodology – but a set of well-known methods are available to apply on the research area and their physical parameters. I fail to see any fruitful discussion on the generated datasets. The scientific problem is analyzed and solved. The introduction part is clear and the scientific problem has clearly identified and addressed. Many thanks for your thorough review of our manuscript. 

My co-authors and I have worked diligently to address each concern raised in your review and we think your feedback has greatly improved the manuscript. 

Regarding your concern of fruitful discussion, we have now included additional text at Lines 426-429: “This is important because most research in this area has focused on suburban green infrastructure. Understanding potential disparities in green infrastructure provision in urban centres is vital to producing strategies that promote socio-ecological equity.”

Lines 462-467: “Additionally, disparities in green infrastructure have important implications for biodiversity conservation efforts. We live in an epoch characterised by a biodiversity crisis, partly due to habitat loss, landscape fragmentation [49,50], and other factors associated with urbanisation such as air and light pollution [51]. Therefore, biodiversity needs enhanced and contiguous ecological, infrastructural, and societal support across the landscape including in urban centres, which can be neglected as our study shows.”

Lines 499-508: “Indeed, although this is pertinent across the board, our study reveals that some urban centres are significantly lacking in health-promoting and biodiversity-supporting green attributes compared to others. Therefore, from an urban centre perspective, we provide an important indication of where green infrastructure support is most needed in GB. This information can potentially be used by the UK government and/or city-level authorities to reduce socio-ecological inequity. For instance, Members of Parliament, urban planners, and campaigners in the lower-ranked urban areas can use our study as an impetus to improve the quality of urban centres in these areas, particularly in the light of the levelling up agenda of the current UK government. The study can also be used as a platform by international researchers to explore potential disparities in urban centre green attributes in other countries.”

2. The authors discuss clear point about the Greenness tree cover but not emphasize on application - they have studied Great Britain area and their MS used the Geospatial application. Thanks for this feedback. As above, we have now placed further emphasis on the potential applications of our findings that disparities in the green infrastructure provision between GB urban centres exist. We have also strengthened the objectives of the study in the Introduction section to further emphasise the importance of the process: Lines 113-120.

i.e., “However, little is known about the equity of green infrastructure provision in city centres. Disparities in green infrastructure could underscore an important socio-ecological justice issue, with some populations gaining the benefits of healthy urban ecosystems and others enduring the disbenefits of poor green infrastructure provision. The same applies to biodiversity––i.e., it is important to understand potential disparities in wildlife-supporting green infrastructure in city centres. Considering these factors, we argue that more emphasis should be placed on mapping and enhancing the green attributes (e.g., trees, broader vegetation cover, and publicly accessible greenspaces such as parks) of urban cores/centres. In doing so, there is considerable potential to provide a range of positive health benefits to many people across the socioeconomic spectrum via the augmented provision of health-promoting green features such as urban forests and recreational greenspaces. There is also potential to enhance biodiversity and interspecies health.”

3. The introduction is good, the method section is trivial and vague at places. Discussion is existing but not extensive. Increase the literature and cite in the introduction part (more than 70 papers cited which is more enough but I find many papers not related to the study work. Please remove extra cited papers and be relevant to the research topic). Thanks for this valuable feedback. We have now augmented the Methods section in response to your comments. We have also provided a methodological workflow diagram (now Figure 2) to increase the clarity of the approaches, decision-making and analytical processes involved in the study. 

We have also included additional citations in the Introduction section with discussion points e.g., Lines 98-106: “Studies assessing the presence and impacts of urban green attributes typically focus on the places where most people reside, such as suburban zones [16,17,18]. Several studies have also used remote sensing-based green cover identification methods in cities. For instance, NDVI is widely used to estimate green cover in urban areas [19,20] and the Enhanced Vegetation Index (EVI) has also been used [21]. NDVI is considered more sensitive to leaf chlorophyll concentrations via the red spectral band (620–670 nm) while EVI more sensitive to canopy structure and leaf area via the near-infrared (NIR) band (840–875 nm) [22]. Ordnance Survey’s (Britain’s national mapping agency) Open Greenspace data is also widely used in urban greenspace studies to explore the distribution of publicly accessible greenspaces [23].”

Lines 113-119: “However, little is known about the equity of green infrastructure provision in city centres. Disparities in green infrastructure could underscore an important socio-ecological justice issue, with some populations gaining the benefits of healthy urban ecosystems and others enduring the disbenefits of poor green infrastructure provision. The same applies to biodiversity––i.e., it is important to understand potential disparities in wildlife-supporting green infrastructure in city centres.”

We have also augmented the Discussion section at Lines 426-429: “This is important because most research in this area has focused on suburban green infrastructure. Understanding potential disparities in green infrastructure provision in urban centres is vital to producing strategies that promote socio-ecological equity.”

Lines 462-467: “Additionally, disparities in green infrastructure have important implications for biodiversity conservation efforts. We live in an epoch characterised by a biodiversity crisis, partly due to habitat loss, landscape fragmentation [49,50], and other factors associated with urbanisation such as air and light pollution [51]. Therefore, biodiversity needs enhanced and contiguous ecological, infrastructural, and societal support across the landscape including in urban centres, which can be neglected as our study shows.”

Lines 499-508: “Indeed, although this is pertinent across the board, our study reveals that some urban centres are significantly lacking in health-promoting and biodiversity-supporting green attributes compared to others. Therefore, from an urban centre perspective, we provide an important indication of where green infrastructure support is most needed in GB. This information can potentially be used by the UK government and/or city-level authorities to reduce socio-ecological inequity. For instance, Members of Parliament, urban planners, and campaigners in the lower-ranked urban areas can use our study as an impetus to improve the quality of urban centres in these areas, particularly in the light of the levelling up agenda of the current UK government. The study can also be used as a platform by international researchers to explore potential disparities in urban centre green attributes in other countries.”

4. I don't feel qualified to judge about the English language but the manuscript style needs improvement according to the authors guidelines. We have double checked the authors guidelines and have spent time enhancing the English language throughout. 

5. Abstracts and Introduction does give a concise information about objective of this work. Why is it needed, what novel is in this research, which papers concerned these research questions to the greenery in the Great Britain, and why? The reasons to study the chosen area in well explained? Why greenery changes in Great Britain region are important (besides urbanization)? Which studies are for this region? Partially, these questions were answered, and in my opinion, it is enough and quite efficiently. Many thanks for this feedback. We are pleased you find the Abstract and Introduction sections sufficient. However, we have taken your “partially” comment on board and have now augmented both sections with additional clarity and detail. 

For instance, in the Abstract, we have reworked Lines 25-36, adding in further context and clarity. We have also included additional text in the Introduction at Lines 113-120 to strengthen the objectives of investigating potential disparities in green infrastructure provision in GB. 

i.e., “However, little is known about the equity of green infrastructure provision in city centres. Disparities in green infrastructure could underscore an important socio-ecological justice issue, with some populations gaining the benefits of healthy urban ecosystems and others enduring the disbenefits of poor green infrastructure provision. The same applies to biodiversity––i.e., it is important to understand potential disparities in wildlife-supporting green infrastructure in city centres. Considering these factors, we argue that more emphasis should be placed on mapping and enhancing the green attributes (e.g., trees, broader vegetation cover, and publicly accessible greenspaces such as parks) of urban cores/centres. In doing so, there is considerable potential to provide a range of positive health benefits to many people across the socioeconomic spectrum via the augmented provision of health-promoting green features such as urban forests and recreational greenspaces. There is also potential to enhance biodiversity and interspecies health.

6. Data source and processing section is completed. Authors can add more information about the datasets acquired (if it feels necessary, software used, processing methods applied to the data: which one is the final resolution as the research was conducted with datasets at different spatial resolutions? A reference is needed for the different indices. – references provided. We have now added a few more details on the methods e.g., see the methods workflow (Figure 2) and additional information between Lines 186-238.The references for the indices used are provided i.e., see citation for i-Tree, Sentinel/EDINA Digimap, and Ordnance Survey (OS) Open Greenspace datasets.

7. Description of methodology is concise. Maybe, the classification technique could be visualized or/and quantified based on more example?

Recommendation. Summarize my comments, Overall, the manuscript is written well. I would recommend to Authors to address my comments into manuscript taking into account most of issues including motivation, methods, results analysis. These are the major revisions and after the complete elaboration, the manuscript could be accepted in this Journal. Thanks. This is a good point. As stated above, we have now augmented the methods sections and have provided a new workflow diagram. 

We have also included a limitations section at Lines 644-657:

“Limitations

The study has several limitations. For instance, the satellite data used within the study was a composite dataset provided through the EDINA Digimap Service. It is therefore possible that geographic disparities may be present, for example, different data timeframes. Such limitations are outside the control of this exploratory study. Some data were not available, for instance, IMD for all countries and green metrics for Northern Ireland (hence being omitted). Other vegetation indices (such as the EVI) could also provide different results in urban areas and should be considered in future studies. The restricted scope of our study (e.g., focusing on GB and sampling urban areas with >100,000 population) means the results should be extrapolated with caution.”

Thanks again for your valuable feedback. We all agree it has greatly improved the manuscript and we hope you now find it in an acceptable form for publication in PLOS ONE.

Reviewer 3 

The Paper presents a comprehensive and detailed evaluation of urban centers' green infrastructure. The Paper has the potential to be published but some essential points that I listed below should be reconsidered or revised. Many thanks for your thorough review of our manuscript. 

My co-authors and I have worked diligently to address each concern raised in your review and we think your feedback has greatly improved the manuscript. 

1. Abstract should support the findings with quantitative analysis results. Thanks. We have now provided additional clarity and context in the Abstract section e.g., at Lines 26-36: “Despite there being a large number of studies investigating greenspace disparities in suburban areas, no known studies have compared the green attributes (e.g., trees, greenness, and greenspaces) of urban centres. Consequently, there may be uncharacterised socioecological disparities between the cores of urban areas (e.g., city centres). This is important because people spend considerable time in urban centres due to employment, retail and leisure opportunities. Therefore, the availability of––and disparities in––green infrastructure in urban centres can affect many lives and potentially underscore a socio-ecological justice issue. To facilitate comparisons between urban centres in Great Britain, we analysed open data of urban centre boundaries with a central business district and population of ≥100,000 (n = 68). Given the various elements that contribute to ‘greenness’, we combine a range of different measurements (trees, greenness, and accessible greenspaces) into a single indicator.”

We have also used some key quantitative outputs e.g., at Lines 42-45: “For instance, Exeter scored the highest with a mean NDVI of 0.15, a tree coverage of 11.67%, and an OS Greenspace coverage of 0.05%, and Glasgow the lowest with a mean NDVI of 0.02, a tree cover of 1.95% and an OS Greenspace coverage of 0.00%.”

2. I believe remote sensing based green cover identification in city centers needs a short paragraph in the Introduction. There are several attempts on this topic, using different methodologies and different vegetation indices, and few of them should be cited. Thanks for this feedback. We have now included a short paragraph on remote sensing based green cover identification in cities and different (not all) vegetation indices at Lines 98-106, with additional citations: 

“Studies assessing the presence and impacts of urban green attributes typically focus on the places where most people reside, such as suburban zones [16,17,18]. Several studies have also used remote sensing-based green cover identification methods in cities. For instance, NDVI is widely used to estimate green cover in urban areas [19,20] and the Enhanced Vegetation Index (EVI) has also been used [21]. NDVI is considered more sensitive to leaf chlorophyll concentrations via the red spectral band (620–670 nm) while EVI more sensitive to canopy structure and leaf area via the near-infrared (NIR) band (840–875 nm) [22]. Ordnance Survey’s (Britain’s national mapping agency) Open Greenspace data is also widely used in urban greenspace studies to explore the distribution of publicly accessible greenspaces [23].”

3. Methods section clearly needs a flowchart including data, implemented methods for parameter extraction, main methodology for decision making, and result representations. This is a great point – thanks.

We have now provided an additional figure with the methodological workflow for greater clarity (now Figure 2) and we agree it strengthens the Methods section. 

4. Authors should explain why they choose a specific indices or method among others in a convincing way (superiority over other ones?) for example why you selected NDVI among other vegetation indices or why you selected PCA? We have now augmented the justification for the NDVI greenness metric in the Methods sections at Lines 219-222: “This ‘greenness’ score has been used as a proxy for vegetation biomass and vegetation cover in other green infrastructure and geospatial studies [34,35], hence being considered suitable for this study (whilst recognising other indices are available).” We acknowledge that other indices are available and may/may not be better depending on various factors and perspectives, however, we chose the NDVI, which has been widely used in urban greenspace studies. 

Additionally, we stated that PCA “was chosen as the preferred method as it accounts for the degree of interrelationship between variables” at Lines 304-306 in the Methods section.

5. In lines 126-133, I do not think you need to give the software and their processing steps but you should explain the algorithm. Thanks for this feedback. We have retained the software information and steps but have taken your comments on board and have now provided additional information on the algorithms between Lines 187-241. 

6. In lines 139-143, again how the data is collected and processed by who is not a concern, but technical details of the satellites and image frame dates are more important. Different seasons will affect the determination of vegetation, thus frame dates should be as closer as much in a country level mosaic. Thanks. Please see above, which includes our response for this comment. We have also stated the dates for the satellite data collection – i.e., a composite from across seasons. We contacted the remote sensing specialists at EDINA Digimap to confirm this information. 

“The Sentinel-2 satellite collected this dataset in 2019, and it was downloaded by the researchers from the EDINA Digimap Service in August 2020 [33]. The Sentinel-2 images used were cloud-free composites collected on various dates and sourced across the calendar year 2019.”

We have also included a limitations section at Lines 644-657 to include information on the satellite data:

“Limitations

The study has several limitations. For instance, the satellite data used within the study was a composite dataset provided through the EDINA Digimap Service. It is therefore possible that geographic disparities may be present, for example, different data timeframes. Such limitations are outside the control of this exploratory study. Some data were not available, for instance, IMD for all countries and green metrics for Northern Ireland (hence being omitted). Other vegetation indices (such as the EVI) could also provide different results in urban areas and should be considered in future studies. The restricted scope of our study (e.g., focusing on GB and sampling urban areas with >100,000 population) means the results should be extrapolated with caution. ”

6. In line 140, not "Sentinal" but "Sentinel". Thanks. We have now updated this.

7. In line 144, what do you mean by isolation, you do not need to separate the bands while calculating NDVI. Thanks. We have now revised this. 

8. In line 147, formulae should be given as NDVI= (Near Infrared- Red)/( Near Infrared +Red) no need for the light term, and NDVI should be on the left-hand side. Thanks. We have taken your feedback on board and have now changed the presentation of the formula.

9. The result of the NDVI is not a direct greenness score. For example, high density vegetation cover with lower chlorophyll content (due to season or other conditions) will provide lower values. Please take advice from a remote sensing professional on this issue. Thanks, we appreciate your thoughts on this. We understand that NDVI is not an absolute greenness metric but an indication of chlorophyll output (whilst detecting noise from other sources). This is a known limitation but we also state that we use the metric as an “estimation of land cover greenness” at Line 217-218. Moreover, the greenness output is significantly correlated (with a strong positive coefficient) with tree cover in our urban areas – please see Appendix I. This provides a relatively good indication that greenness in our boundaries is largely detected from the tree coverage in the urban centres. 

11. Lines 166-172 needs rewriting for clear explanation and again more focus on the algorithm. Thanks. We have now revised this section and provided additional detail on the algorithm. 

12. I highly encourage not to use mean NDVI directly as it will be affected by the distribution of land use classes and may not reflect the greenness of a city center. It is well observed in Figure 3. that Exeter has quite more green areas with respect to total cover but takes the same value as Cambridge or Sutton coldfield. Also, Basildon does not have that less green cover to take 0.01. I recommend applying a threshold for green cover, calculate the area of green cover, and ratio it according to the total area of that city boundary. Thanks for this feedback. Although we appreciate your comments, we would like to retain our approach and outputs in this instance. As above, we argue that as the greenness output is significantly correlated (and strongly) with tree cover in our urban areas, our results provide a good indication that greenness in our boundaries is largely detected from the tree coverage in the urban centres and the greenness scores are therefore consistent and robust. 

We have, however, added the following to the Discussion to reflect your points:

“Future research could replicate the work adding additional analysis, for example, through using NDVI thresholds to identify green areas rather than applying mean NDVI values.”

13. Figure qualities are low in general. Thanks. We have checked the figures and have increased the resolution to at least 400 dpi (standard journals request a minimum of 300 dpi).

14. In discussion, please avoid long writings that belongs to other studies but not matching your findings directly (e.g., lines 329-338). Thanks for your feedback. We have checked through the discussion and have retained some of the discussion points as we think it provides important context regarding the features (biodiversity) and potential benefits (socio-ecological). However, we have taken your comments on board and have included additional discussion points regarding what our findings show to provide greater focus. 

We have included additional text at Lines 426-429: “This is important because most research in this area has focused on suburban green infrastructure. Understanding potential disparities in green infrastructure provision in urban centres is vital to producing strategies that promote socio-ecological equity.”

Lines 462-467: “Additionally, disparities in green infrastructure have important implications for biodiversity conservation efforts. We live in an epoch characterised by a biodiversity crisis, partly due to habitat loss, landscape fragmentation [49,50], and other factors associated with urbanisation such as air and light pollution [51]. Therefore, biodiversity needs enhanced and contiguous ecological, infrastructural, and societal support across the landscape including in urban centres, which can be neglected as our study shows.”

Lines 499-508: “Indeed, although this is pertinent across the board, our study reveals that some urban centres are significantly lacking in health-promoting and biodiversity-supporting green attributes compared to others. Therefore, from an urban centre perspective, we provide an important indication of where green infrastructure support is most needed in GB. This information can potentially be used by the UK government and/or city-level authorities to reduce socio-ecological inequity. For instance, Members of Parliament, urban planners, and campaigners in the lower-ranked urban areas can use our study as an impetus to improve the quality of urban centres in these areas, particularly in the light of the levelling up agenda of the current UK government. The study can also be used as a platform by international researchers to explore potential disparities in urban centre green attributes in other countries.”

15. Lastly and importantly, I would like to see maps of your findings. GIS tools can help you easily to perform these tasks. Thanks for this comment. We feel that we have provided sufficient figure outputs for this study, showing the relevant results and telling a coherent story (which has been greatly improved as a result of your feedback). We have already provided some outputs in map-form e.g., Figure 1 (showing distribution of sample locations) and Figure 4 (showing NDVI outputs) and feel that given the type of data used and its distribution and scale, further map-style plots would fail to be visually impactful/helpful. 

Thanks again for your valuable feedback. We all agree it has greatly improved the manuscript and we hope you now find it in an acceptable form for publication in PLOS ONE.

Reviewer 4 

The authors have applied geospatial techniques to compare green infrastructure among urban centers in Great Britain. This study is conducted with the aim that it helps stakeholders to make decisions regarding green infrastructure. The abstract is informative and the reference list captures all the citations in the text. However, a few corrections are needed. Many thanks for your thorough review of our manuscript. 

We are pleased you see value in the manuscript.

My co-authors and I have worked diligently to address each concern raised in your review and we think your feedback has greatly improved the manuscript. 

The authors should proofread the entire manuscript to check for grammatical and typo errors. For instance “Sentinal-2” on line 140 should be “Sentinel-2”. Thanks. This has now been updated. 

On line 93, the sentence should read (Northern Ireland was excluded due to data unavailability). As above. This has now been updated.

In Figure 3, the authors omitted a scale and north arrow. As above. This has now been updated.

Thanks again for your valuable feedback. We all agree it has greatly improved the manuscript and we hope you now find it in an acceptable form for publication in PLOS ONE.

---

## [Decision Letter · Decision Letter 1]

6 Oct 2022

PONE-D-21-34255R1Urban centre green metrics in Great Britain: A geospatial and socioecological studyPLOS ONE

Dear Dr. Robinson,

Thank you for submitting your manuscript to PLOS ONE. After careful consideration, we feel that it has merit but does not fully meet PLOS ONE’s publication criteria as it currently stands. Therefore, we invite you to submit a revised version of the manuscript that addresses the points raised during the review process.

As recommended by the reviewers, there are only minor issues that need to be revised and these are given in reviewers' reports. 

We look forward to receiving your revised manuscript.

Kind regards,

Eda Ustaoglu, PhD

Academic Editor

PLOS ONE

Journal Requirements:

Reviewers' comments:

Reviewer's Responses to Questions

**Comments to the Author**

1. If the authors have adequately addressed your comments raised in a previous round of review and you feel that this manuscript is now acceptable for publication, you may indicate that here to bypass the “Comments to the Author” section, enter your conflict of interest statement in the “Confidential to Editor” section, and submit your "Accept" recommendation.

Reviewer #1: All comments have been addressed

Reviewer #3: All comments have been addressed

Reviewer #4: All comments have been addressed

2. Is the manuscript technically sound, and do the data support the conclusions?

Reviewer #1: Yes

Reviewer #3: Partly

Reviewer #4: (No Response)

3. Has the statistical analysis been performed appropriately and rigorously? 

Reviewer #1: Yes

Reviewer #3: Yes

Reviewer #4: (No Response)

4. Have the authors made all data underlying the findings in their manuscript fully available?

Reviewer #1: Yes

Reviewer #3: Yes

Reviewer #4: (No Response)

5. Is the manuscript presented in an intelligible fashion and written in standard English?

Reviewer #1: Yes

Reviewer #3: Yes

Reviewer #4: (No Response)

6. Review Comments to the Author

Reviewer #1: Urban centre green metrics in Great Britain: A geospatial and socioecological study

Most of my comments have been addressed.

Minor issues:

1.There are a lot of repetitive keywords (e.g., urban green infrastructure and green infrastructure).

2.The data was collected in 2019, so how did the authors draw a conclusion related to the COVID-19 pandemic?

Reviewer #3: In revised version I can see that most of my recommendations are performed and some of them are stated to be limitation i the paper. Now it can be considered for publication however the quality of the figures is still an issue. Authors stated they increased the dpi however, some figures become even worse (please check the pdf of your submission). Maybe it is the problem that images are exported in small dimensions (height x width in terms of pixel size). nevertheless this is a technical issue that maybe journal guide Authors.

Reviewer #4: (No Response)

7. PLOS authors have the option to publish the peer review history of their article (what does this mean?). If published, this will include your full peer review and any attached files.

Reviewer #1: No

Reviewer #3: No

Reviewer #4: No

---

## [Author Response · Author response to Decision Letter 1]

7 Oct 2022

Editor

Thanks again for considering our manuscript for PLOS ONE. We are pleased that you and the reviewers all see the value of our manuscript.

My co-authors and I are also pleased that we addressed the major revisions and that the reviewers feel the manuscript is now ready for publication. We have also addressed the minor revisions from the latest round of feedback and agree that the reviews greatly improved the manuscript. We hope you now find it in an acceptable form for publication in PLOS ONE.

Reviewer #1

Thanks for your feedback. To address this, we have removed some of the repetitive keywords in particular “green infrastructure”; for instance, at Lines 91, 94, 361, 363, 377, 378, 410, 552. 

Thanks. We have taken your feedback on board and have removed one of the two references to COVID-19 in the discussion (at Line 555). The pandemic is a topical and a highly relevant to current urban centres and future green infrastructure management. Therefore, we left one reference to COVID-19 in the manuscript (as a relevant but not central discussion point), at Line 398: “Urban centres in Great Britain are changing, and retail outlets are closing, mainly due to the evolution of digital shopping technologies. This has been accentuated by the COVID-19 pandemic, which has devasted the traditional retail property sector”. 

Reviewer #3

We are pleased you feel the paper is now ready for publication and appreciate you spending time to review it.

Thanks for pointing the image issue out. Perhaps it is a conversion issue; we will double check the quality and upload new versions. We will also liaise with the journal editors to make sure they are at the appropriate resolution/quality to publish.

---

## [Decision Letter · Decision Letter 2]

18 Oct 2022

Urban centre green metrics in Great Britain: A geospatial and socioecological study

PONE-D-21-34255R2

Dear Dr. Robinson,

We’re pleased to inform you that your manuscript has been judged scientifically suitable for publication and will be formally accepted for publication once it meets all outstanding technical requirements.

Kind regards,

Eda Ustaoglu, PhD

Academic Editor

PLOS ONE

Additional Editor Comments (optional):

Reviewers' comments:

Reviewer's Responses to Questions

**Comments to the Author**

1. If the authors have adequately addressed your comments raised in a previous round of review and you feel that this manuscript is now acceptable for publication, you may indicate that here to bypass the “Comments to the Author” section, enter your conflict of interest statement in the “Confidential to Editor” section, and submit your "Accept" recommendation.

Reviewer #1: All comments have been addressed

Reviewer #3: All comments have been addressed

2. Is the manuscript technically sound, and do the data support the conclusions?

Reviewer #1: Yes

Reviewer #3: Yes

3. Has the statistical analysis been performed appropriately and rigorously? 

Reviewer #1: Yes

Reviewer #3: Yes

4. Have the authors made all data underlying the findings in their manuscript fully available?

Reviewer #1: Yes

Reviewer #3: Yes

5. Is the manuscript presented in an intelligible fashion and written in standard English?

Reviewer #1: Yes

Reviewer #3: Yes

6. Review Comments to the Author

Reviewer #1: Urban centre green metrics in Great Britain: A geospatial and socioecological study

I have no further comments.

Reviewer #3: Second revision of the paper satisfies my my minor concerns and now the paper seems suitable for publish.

7. PLOS authors have the option to publish the peer review history of their article (what does this mean?). If published, this will include your full peer review and any attached files.

Reviewer #1: No

Reviewer #3: No

---

## [Editor Report · Acceptance letter]

28 Oct 2022

PONE-D-21-34255R2 

Urban centre green metrics in Great Britain: A geospatial and socioecological study 

Dear Dr. Robinson:

I'm pleased to inform you that your manuscript has been deemed suitable for publication in PLOS ONE. Congratulations! Your manuscript is now with our production department. 

Kind regards, 

on behalf of

Dr. Eda Ustaoglu 

Academic Editor

PLOS ONE